# Experience of Euclidean geometry sculpts the development and dynamics of rodent hippocampal sequential cell assemblies

Usman Farooq [1] & George Dragoi [1,2,3] ✉

Euclidean space is the fabric of the world we live in. Whether and how geometric experience shapes our spatial-temporal representations of the world remained unknown. We deprived male rats of experience with crucial features of Euclidean geometry by rearing them inside spheres, and compared activity of large hippocampal neuronal ensembles during navigation and sleep with that of cuboid cage-reared controls. Sphere-rearing from birth permitted emergence of accurate neuronal ensemble spatial codes and preconfigured and plastic time-compressed neuronal sequences. However, sphere-rearing led to diminished individual place cell tuning, more similar neuronal mapping of different track ends/corners, and impaired pattern separation and plasticity of multiple linear tracks, coupled with reduced preconfigured sleep network repertoires. Subsequent experience with multiple linear environments over four days largely reversed these effects. Thus, early-life experience with Euclidean geometry enriches the hippocampal repertoire of preconfigured neuronal patterns selected toward unique representation and discrimination of multiple linear environments.

Spontaneous neuronal activity of the hippocampus embodies an internal model of external space[1–5], which could support its critical role in navigation[2,6] and contextual learning[7,8]. The hippocampal network of immature experimentally-naïve rats spontaneously expresses time-compressed preconfigured preplay sequence motifs during sleep[9,10] that are recruited to encode future de-novo sequential experiences on linear tracks as early as postnatal day 17 (P17) and future discrete spatial locations as early as P15[11]. This suggests that the internal model of linear space is largely innate[12] and could emerge likely via genetically-encoded developmental programs[13] in the absence of explicit navigational experience once synaptic wiring matures[11,14]. However, owing to the geometry of the spatial environment in which they were raised (i.e., a cuboid-shaped home cage), these rats were inherently exposed to linear boundaries, vertical borders, planar floors, and right-angled corners, which are crucial features of Euclidean space. These and other properties of Euclidean space facilitate sequentialization, segmentation, and orthogonalization of spatial experiences providing direct experience with geometric linearity within Euclidean space, which we refer to as 'geometric linearity'. This daily 'unaccounted' exposure to geometric linearity during early life may have enabled the formation of an internal model for linear space in lieu of or in addition to intrinsic developmental factors and non-spatial experiences as a form of latent learning[6]. To determine the contributions of early-life latent learning in Euclidean space and of age-related intrinsic developmental programs and non-spatial experiences toward the emergence of an internal, preconfigured model of space, we deprived rats of experience with geometric linearity and compared the neurodevelopment of hippocampal neuronal ensembles in deprived and normally-reared rats. The deprived rats were raised from birth together with their littermates and the dam inside 23-inch diameter bedded, externally lighted, white translucent plastic sphere, which is one model of non-Euclidean geometry.

Considering that linear experience-related neuronal sequences critical for navigation and memory formation, adult-like hippocampal

---

[1]Department of Psychiatry, Yale School of Medicine, New Haven, CT, USA. [2]Department of Neuroscience, Yale School of Medicine, New Haven, CT, USA. [3]Wu Tsai Institute, Yale University, New Haven, CT, USA. ✉e-mail: george.dragoi@yale.edu

theta sequences[15] and plasticity in replay[16–19] emerge in concert at P23-24[11], we deprived our rats of experience with geometric linearity until P24. We recorded the activity of large ensembles of dorsal CA1 neurons as experimentally-naïve P24-27 sphere-reared rats slept in spheres and ran on multiple linear tracks[10] over four consecutive days. We compared their activity across various brain states (active run behavior, awake rest and sleep) and at various levels of network organization with that of age-matched, younger, and adult cuboid cage-reared control rats (reared in experimenter-controlled environments, see Methods). Finally, our multi-day neuronal ensemble recordings from the same groups of rats allowed us to investigate whether experience of geometric linearity in the form of multiple linear tracks gathered over several days could reshape the hippocampal internal model of space in the sphere-reared rats.

## Results

### Hippocampal neuronal ensembles represent de-novo linear trajectories during run after sphere-rearing

The spherical environment deprived the rats of experience with (1) linear borders and linear trajectories along them, (2) flat surfaces which permit linear trajectories, (3) right-angled corners and vertical geometric landmarks that orthogonalize and segment linear trajectories in the horizontal plane (Fig. 1a, top). Instead, the spherical home cage provided experience with continuously curved borders and concave surfaces (Fig. 1a, bottom). Sphere- and cuboid-reared rats (Fig. 1a, Supplementary Fig. 1a) were implanted bilaterally with Neuronexus silicon probes (total of 64–128 recording sites) over dorsal CA1 area of the hippocampus[11] at P20-21 and returned to their respective home cages. Over the next 2–3 days, the electrodes were gradually moved into the CA1 pyramidal layer while the rats were resting and sleeping in small, heated sleep boxes of corresponding geometry that continued to deprive the sphere-reared rats of physical and visual exposure to geometric linearity (Methods). Activity from CA1 neuronal ensembles and local field potentials were recorded (Fig. 1b) at P23-24 on Day1 while the naïve animals (1) slept in the box for ~90 min (Pre-Day1Run1 sleep), (2) explored for the first time a 1 m-long linear track in the same room for ~30–45 min (de-novo Day1Run1), (3) slept in the box for ~90 min (Post-Day1Run1 sleep), and (4) ran on the same track for ~30 min (Day1Run2). Sphere-rearing did not influence locomotor dynamics during development, as evidenced by similar individual rat locomotion when co-housed with the littermates and mother in rearing contexts of corresponding geometry, tested at different developmental ages (Supplementary Fig. 1b, $N = 7$ cuboid-reared and $N = 8$ sphere-reared rats) using multi-animal DeepLabCut for position estimation[20]. Furthermore, the geometry of the rearing environment did not influence the rats' velocity and overall behavior on linear tracks assessed by the number of traversed laps and head sweeps during running (Fig. 1c, Supplementary Fig. 1c, d). In addition, sphere-rearing did not increase stress levels, assessed by quantifying fecal corticosterone (FC) levels of individual rats that provided a temporally-integrated non-invasive measure of chronic stress[21] in cuboid- and sphere-reared rats exposed to linear tracks (using FC enzyme-linked immunosorbent assay, ELISA: cuboid vs. sphere, $54.0 \pm 7.5$ vs. $44.4 \pm 3.5$ ng/g, $p = 0.26$, two-sided $t$-test, $N = 10$ rats/group, Supplementary Fig. 1e). Therefore, sphere-rearing was a non-stressful protocol to probe the influence of early-life experience with geometric linearity (and lack thereof) on neurodevelopment without impacting locomotor dynamics.

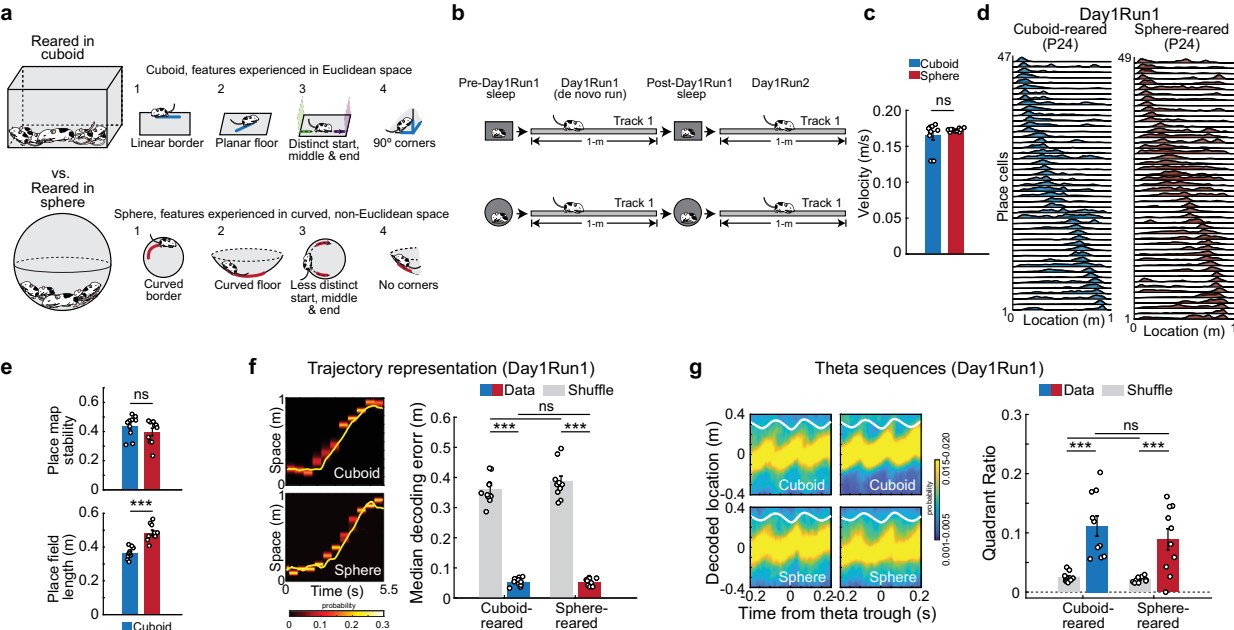

**Fig. 1 | Experimental design and early-life emergence of place cells and theta sequences after sphere-rearing. a** Experimental design: rearing in cuboid (top) or spherical (bottom) cages (left panel), and the critical features of Euclidean geometry experienced in cuboid-cages (blue lines, a1-4, top) and their curved non-Euclidean analogs during deprivation from Euclidean geometry inside spheres (red lines, a1-4, bottom). **b** Experimental design (Day1): Sleep and navigational run sessions on a linear track. **c** Animal velocity during run ($p = 0.79$, two-sided rank-sum test). **d** Examples of simultaneously-recorded place cell sequences in cuboid- (left) and sphere-reared rats (right). **e** Place map stability (top, $p = 0.21$, two-sided rank-sum test) and primary place field length (bottom, $p = 0.00025$, two-sided rank-sum test) during Day1Run1 across experimental groups. **f** Decoded (heatmaps) and real (yellow lines) single-lap Day1Run1 trajectories (left). Decoding error (right) across groups ($p = 0.97$, two-sided rank-sum tests) and compared to shuffles ($p = 0.00098$ for each group, one-sided signed-rank tests). **g** Theta sequences (left, examples) compared across groups ($p = 0.38$, two-sided $t$-test) and against shuffles (cuboid: $p = 0.00037$, sphere: $p = 0.0018$, paired one-sided $t$-tests) using the quadrant ratio method (right). Data in **c**, **e–g** are represented as mean ± SEM. $N = 10$ (5 rats/group; 2 directions/rat). ***$p < 0.005$. ns = not significant. Source data are provided as a Source Data file. Cartoons in Fig. 1/Panels **a**, **b** adapted from U. Farooq, G. Dragoi, Emergence of preconfigured and plastic time-compressed sequences in early postnatal development. Science 363, 168–173 (2019). DOI: 10.1126/science.aav0502. Reprinted with permission from AAAS.

We next simultaneously recorded from an average of 63.0 ± 9.9 (range: 44−99 well-isolated stable CA1 neurons) and 67.6 ± 14.3 neurons (range: 31−111 well-isolated, stable CA1 neurons) across all sleep and run sessions in P23-24 cuboid-reared ($N$ = 5, one P23 and four P24) and P24 sphere-reared ($N$ = 5) rats, respectively (Fig. 1d, Supplementary Fig. 1f). Place cells were expressed in the sphere-reared rats during their first-ever, de novo run on a linear track and maintained a similar degree of place field stability, peak firing rates and directional coding within the run session as the cuboid-reared rats (Fig. 1e, top, Supplementary Fig. 1g, h). This indicates sphere-rearing did not prevent these rats from rapidly expressing a stable representation of the linear space at P24, despite lacking prior experience of this sort. The rapid stable expression of place maps is in contrast with the gradual age-dependent postnatal development of these features in younger P15-P22 cuboid-reared rats[11]. However, sphere-reared P24 rats expressed larger primary place field lengths and lower spatial information for linear space compared with P23-24 cuboid-reared rats (Fig. 1e, bottom, Supplementary Fig. 1i), due to lack of early-life experience with critical features of linear geometry (Fig. 1a). Despite this difference in individual place cell tuning, animals' linear trajectories were represented with similarly high precision by ensembles of place cells across groups using Bayesian decoding at the behavioral timescale[10,11,22] (Fig. 1f; cuboid vs. sphere: 5.3 ± 0.4 vs. 5.2 ± 0.4 cm trajectory decoding median error, $p$ = 0.97, two-sided rank-sum test).

Hippocampal theta sequences, which bind recent-past, present, and immediate-future visited track locations in temporally-compressed manner within individual cycles of theta oscillation (5−9 Hz) during locomotion[15,23], emerge at P23-24 in cuboid-reared rats[11]. In P24 sphere-reared rats, we observed similar theta oscillation frequency during run (6.7 ± 0.2 vs. 6.8 ± 0.1, cuboid vs. sphere-

reared, $p$ = 0.91, two-sided rank-sum test, Supplementary Fig. 1j) and similar theta sequence compression and spatial binding within theta oscillations while rats traversed the middle portion of the track (theta sequence quadrant ratio: 0.11 ± 0.02 vs. 0.09 ± 0.02, cuboid vs. sphere-reared, $p$ = 0.38, two-sided $t$-test; Fig. 1g, Supplementary Fig. 1k−n, for additional quantification of theta sequences using different temporal resolution and methodology). Our results indicate that the emergence of hippocampal neuronal ensembles' competence to accurately represent animal position and linear trajectory at both behavioral timescale and temporally-compressed theta scale during de novo run[11,15,24] does not require prior experience with geometric linearity.

### Hippocampal neuronal ensembles depict de-novo linear trajectories during sleep and awake rest after sphere-rearing

During sleep and rest, neuronal sequences replay experienced[19,25,26] and preplay future novel[9,10] extended linear animal trajectories in a time-compressed manner. P/replay (i.e., preplay and replay) occurs within brief epochs (100−800 ms) of elevated population neuronal activity often associated with high-frequency ripple oscillations (140−250 Hz) called frames[10]. P23-24 cuboid-reared animals naïve to extended linear space exhibited significant preplay and replay of the de-novo Day1Run1 trajectory during sleep and rest frames (Fig. 2a) and experience-related plasticity expressed as stronger trajectory depiction in replay compared with preplay during sleep[11] (Fig. 2c), at similar proportions to adult rats[10,11,16,27,28]. Despite deprivation from prior experience with geometric linearity, P24 sphere-reared animals displayed robust extended linear preplay during Pre-Day1Run1 sleep and replay in Post-Day1Run1 sleep (Fig. 2b, c, Supplementary Fig. 2a−d for supporting p/replay

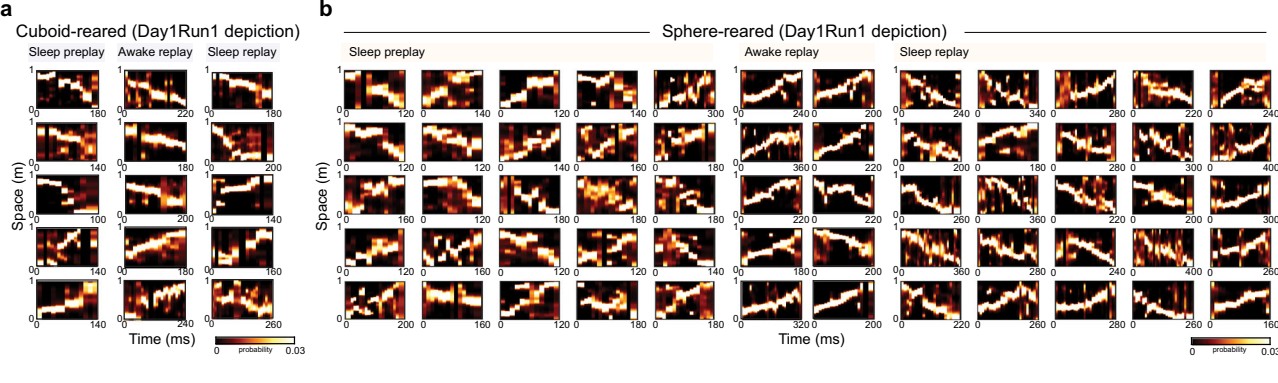

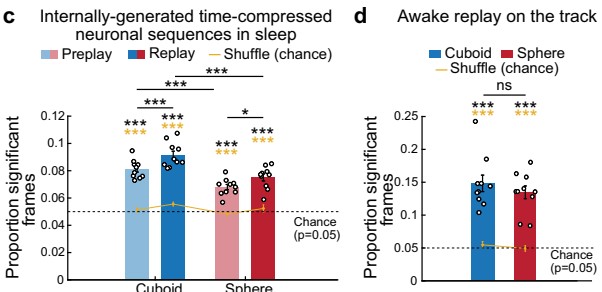

**Fig. 2 | Early-life emergence of sleep preplay, awake replay and plasticity in sleep replay after sphere-rearing.** Examples of Day1: sleep preplay, on-track awake replay, and plasticity in sleep replay depicting Day1Run1 trajectories in cuboid- (**a**) and sphere-reared rats (**b**). **c** Significant incidence of sleep p/replay depicting Day1Run1 in both groups (cuboid and sphere p/replay: vs. shuffle: $p$ = 1.8 × 10⁻⁷, and $p$ = 1.4 × 10⁻⁷, $p$ = 8.7 × 10⁻⁶ and $p$ = 1.1 × 10⁻⁴, paired one-sided $t$-tests, yellow stars; $p$ = 6.9 × 10⁻⁸, and $p$ = 8.5 × 10⁻⁸, $p$ = 3.1 × 10⁻⁶ and $p$ = 2.2 × 10⁻⁶ vs. $p$ = 0.05 chance, one-sided $t$-tests, black stars; $p$ = 0, one-sided Binomial tests vs. $p$ = 0.05 chance for all groups and sleeps). Reduced incidence of p/replay in sphere-

reared rats ($p$ = 2.4 × 10⁻⁴ and $p$ = 5.3 × 10⁻⁴, two-sided $t$-tests). Plasticity in replay (replay > preplay) in both groups (cuboid: $p$ = 0.0019, sphere: $p$ = 0.029, paired two-sided $t$-tests). **d** Incidence of on-track awake replay (cuboid and sphere replay: vs. shuffle: $p$ = 6.5 × 10⁻⁵ and $p$ = 7.1 × 10⁻⁶ paired one-sided $t$-tests, yellow stars; vs. $p$ = 0.05 chance: $p$ = 1.3 × 10⁻⁵ and $p$ = 5.4 × 10⁻⁶ one-sided $t$-tests, black stars; $p$ = 0, one-sided Binomial tests vs. $p$ = 0.05 chance for both groups) is similar across groups ($p$ = 0.40, two-sided $t$-test). Data in **c**–**d** are represented as mean ± SEM. $N$ = 10 (5 rats/group; 2 directions/rat). ***$p$ < 0.005. *$p$ < 0.05. ns = not significant. Source data are provided as a Source Data file.

analyses). This demonstrates that the emergence of network pre-configuration into temporally-compressed neuronal sequences that preplay a future novel extended linear trajectory in experimentally-naïve animals does not require prior experience with geometric linearity.

Whereas sphere-reared animals had lower incidences of preplay and replay compared with their cuboid-reared counterparts, these levels of preplay were sufficient to support experience-related plasticity in neuronal sequence replay[11,16,18] during Post-Day1Run1 sleep (Fig. 2c). Single-cell spatial tuning differences were not sufficient to explain the differences in p/replay across groups (Supplementary Fig. 2e, f). During sleep, ripple power (140–250 Hz) was similar across groups and frames co-occurring with ripples exhibited significant preplay, replay and plasticity in replay in both groups (Supplementary Fig. 2g, h). Consistently, multi-neuronal cell assemblies co-active at 20-ms timescale during run[16,29–31] showed similar levels of plasticity during sleep reactivation in both groups (Supplementary Fig. 2i). Awake-rest replay, time-compressed representation of experienced trajectories during immobility on the track[25] important for spatial planning and stabilization of cognitive maps[32–34], occurred with similar incidence in the cuboid- and sphere-reared rats during immobility epochs of the de novo Day1-Run1 (Fig. 2a, b, d).

## Increased similarity in hippocampal ensemble depiction of distinct track ends/corners alters representation of linear space after sphere-rearing

Sphere-rearing deprived the rats from birth of experience with geometric linearity (including environmental ends/corners and vertical borders) replacing it by experience with curvature, which attenuated the incidence of time-compressed linear p/replay (Fig. 2). We asked whether this affected the network ability, at any level, to reliably map individual locations along an extended linear space. We computed the similarity between the neuronal ensembles mapping pairs of locations along the linear track during Day1Run1. In cuboid-reared rats, locations furthest apart (near the two track ends) recruited distinct neuronal ensembles exhibiting low population vector correlations (Fig. 3a, b).

In sphere-reared rats, locations near the opposite track-ends recruited similar place cell ensembles at P24 compared with cuboid-reared rats (Fig. 3b). This was not due to deficits in directional coding, which developed normally (Supplementary Fig. 1h), anticipation of reward provided at track ends, as it was also evident while rats moved away from the reward-zone (Supplementary Fig. 3a), behavioral biases at track ends (i.e., velocity and track occupancy, Supplementary Fig. 3b) or differences in place cell tuning (Supplementary Fig. 3c).

The representation of continuous linear space after sphere-rearing was affected beyond the track ends. Visualization of the

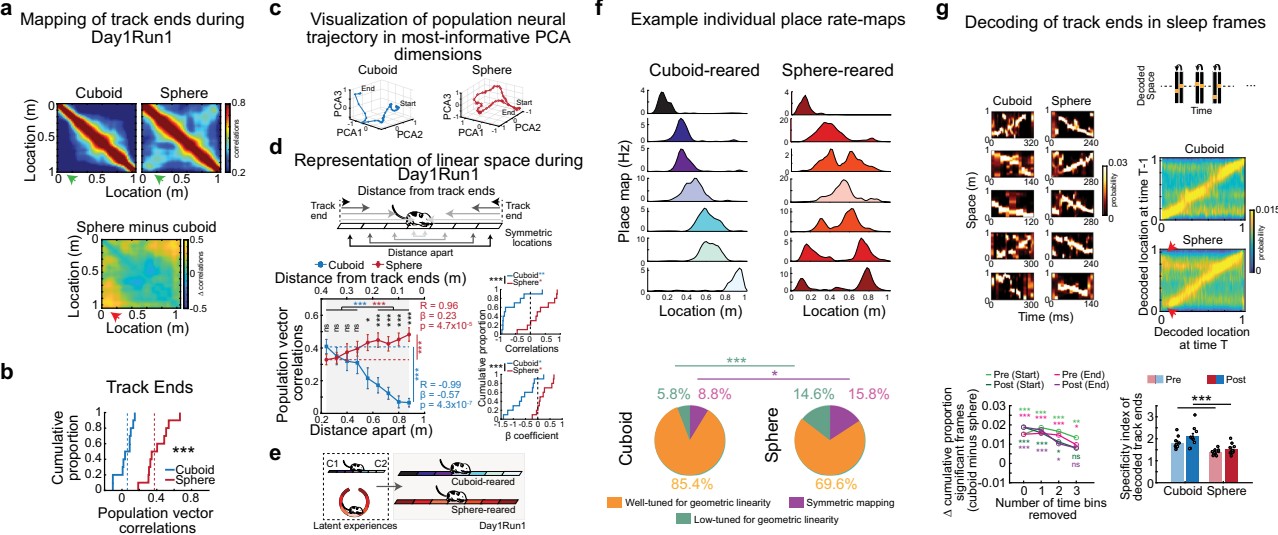

**Fig. 3 | Changes in hippocampal ensemble depiction of Day1Run1 linear track ends/corners after sphere-rearing. a** Group-average correlation matrix depicting neuronal ensemble mapping of track locations for cuboid (top-left), sphere-reared rats (top-right), and difference between groups (bottom). Green/red arrows: higher similarity of population vectors representing track ends in sphere-reared rats. **b** Higher population vector correlation of place maps between the two track ends in sphere- vs. cuboid-reared rats ($p = 0.0018$, two-sided rank-sum tests). Dashed lines: medians. **c** Neural manifold for track across 3 most informative principal component dimensions. **d** Schematic (top) and values (bottom-left) of population vector correlations between place maps at locations equidistant/symmetric from the two ends of the track (and track middle) vs. distance between the locations during Day1Run1 (Pearson's correlation (R); best-fit regression line slope (ß). Stars: significant differences between groups, between closest vs. furthest location, 0.24 vs. 0.88 m (right), between four adjacent spatial locations in middle vs. end (towards top). Dashed lines: group-averages at 0.24 m distance. Bottom-right: Correlations (top) and beta-coefficients (ß, slope) of best-fit regression line (bottom) for all individual animals and run directions (colored stars: comparison vs. 0). **e** Increased similarity of neuronal ensembles active at track ends/corners and its gradual decrease between symmetric track locations as a function of distance from ends in sphere-reared rats, reflecting early latent experiences in cuboidal (line) or spherical (ring) home cages. **f** (Top) Example individual run direction place rate maps with

non-symmetric and symmetric firing in cuboid- (left) and sphere-reared rats (right; rate maps 3, 5, 6, 7 counting top to bottom exhibit some symmetry). (Bottom) Proportion rate maps well-tuned for linear space, exhibiting symmetric track-firing and low tuning for linear space. Symmetric track-firing and low tuning occurred more frequently in sphere-reared rats ($p = 0.018$ and $p = 0.0015$, respectively, two-sided Z-tests for 2 proportions). **g** Left-top two columns: Example Day1Run1 decoded trajectories during Post-Day1Run1 sleep frames in cuboid (left) and sphere-reared rats (right) illustrating confusion/swapping of decoded probabilities of track ends after sphere-rearing. Left-bottom: Reduced differences in cumulative significant frame proportions between cuboid and sphere-reared rats during Pre-/Post-Day1Run1 sleep on removal of 1–3 time bins (20 ms/bin) from beginning or end of non-significant sleep frames. Right-top: Method for determining time-space contingency. Right-center: Average time-space contingency matrices of decoded locations along the linear track; arrowheads: swapping of decoding of track ends in sphere-reared rats. Right-bottom: Specificity index of decoded track ends ($p = 0.00050$, $p = 0.0025$, two-sided $t$-tests). Data in **d**, **g**, are means ± SEM. $N = 10$ (5 rats/group; 2 directions/rat). ***$p < 0.005$. **$p < 0.01$. *$p < 0.05$. ns = not significant. Source data provided as Source Data file. Cartoons in Fig. 3/Panels d, e adapted from U. Farooq, G. Dragoi, Emergence of preconfigured and plastic time-compressed sequences in early postnatal development. Science 363, 168–173 (2019). DOI: 10.1126/science.aav0502. Reprinted with permission from AAAS.

neural 'trajectory' on the 1-m track taken by the place cell ensembles in the top 3 principal component dimensions indicated a higher curvature after sphere-rearing (Fig. 3c). For quantification, we compared population vector correlations between place maps in the non-reduced data at equidistant locations on the linear track using the two track ends as references (or the track middle, as an alternate reference frame) (Fig. 3d, top). In P23-24 cuboid-reared animals, the ensemble representation of locations further apart in linear space during de novo Day1Run1 was more distinct than of those nearby (Fig. 3d, bottom-left), allowing spatial mapping/discrimination. In contrast, sphere-reared rats had a significant positive relationship between the distance separating equidistant track locations and the similarity of recruited neuronal ensembles due to significantly higher population vector correlation values for locations further apart on the track (Fig. 3d, bottom-left, cuboid vs. sphere comparison for locations 0.24 to 0.88 m apart, $p = 0.18, 0.82, 0.92, 0.44, 0.011, 0.0013, 4.1 \times 10^{-5}, 9.1 \times 10^{-8}, 1.2 \times 10^{-7}$ respectively, two-sided rank-sum tests; closest vs. furthest location, 0.24 vs. 0.88 m, cuboid: $p = 8.9 \times 10^{-5}$, sphere: $p = 0.0032$, two-sided signed-rank tests, stars on right of Fig. 3d bottom-left; four spatial locations in middle vs. four spatial locations at ends, cuboid: $p = 4.6 \times 10^{-10}$, sphere: $p = 3.8 \times 10^{-4}$, two-sided rank-sum tests, stars towards top of Fig. 3d bottom-left). These differences in representation of linear space were observed at the individual animal and run direction level as well (Fig. 3d, bottom-right; cuboid- vs. sphere-reared correlations and beta-coefficients, $p = 0.0017$ and $p = 0.0022$, two-sided rank-sum tests; cuboid: correlations < 0, $p = 0.0098$ and beta-coefficients < 0, $p = 0.014$, sphere: correlations > 0, $p = 0.049$ and beta-coefficients > 0, $p = 0.049$, two-sided signed-rank tests). We propose these differences in representation of linear space are due to differences in early-life geometric experience (depicted as a cartoon in Fig. 3e). To determine if this was due to hippocampal network immaturity after sphere rearing, we investigated 16 naïve younger cuboid-reared rats between ages P15-22 undergoing the same Day1Run1 protocol (Methods). Unlike the sphere-reared rats, the similarity of neuro-ensembles recruited at equidistant locations decayed as a function of distance between the locations as early as P15 (Supplementary Fig. 4a). The altered representation after sphere-rearing was not due to place cell tuning differences (i.e., spatial information and primary place field lengths, Supplementary Fig. 4b). Notably, locations were represented sufficiently differently in sphere-reared rats to enable an accurate behavioral timescale representation of the track (Fig. 1f). Evidence of an altered representation was also observed at the individual place rate-map level, with a small, significant increase in proportion of rate maps exhibiting less tuning or symmetric mapping of equidistant locations after sphere-rearing (Fig. 3f), in agreement with our neuronal ensemble-level results (Fig. 3a–e). Only ~10% of cells with symmetric rate maps for a run direction exhibited symmetric maps for the other run direction in the sphere-reared rats, which were not significantly different from the cuboid-reared rats (p = 0.16, two-sided Z-test for two-proportions). This suggests that this altered representation might be a neuronal ensemble phenomenon rather than an inherent property of individual cells within the network.

As place cell sequences for linear space are p/replayed during the preceding/following sleep[5,9,11,19], and linear p/replay was relatively reduced after sphere-rearing, we further investigated the effect of experience with non-linear, curved space on the decoded sleep trajectories. We focused on the depiction of the two track ends given that these were more likely to recruit similar ensembles during run after sphere-rearing (Fig. 3b). We observed a network 'confusion' of the two track ends in a small subgroup of decoded sleep frames in the sphere-reared rats (selected example frames in Fig. 3g, left) consisting of a co-representation or 'swapping' of the two track ends (or that of other symmetric locations near the track ends) along with an accurate depiction of the remaining trajectory. Multiple analyses supported this observation. First, since the track-end (or other symmetric locations)

swapping/confusion was most apparent in the sleep frames' start or end, for non-significant frames we removed 20–60 ms-long epochs (1–3 time bins, 20 ms/bin) from their start or end and quantified the sequential trajectory content in the resulting reduced frames (as in Fig. 2). Since decoded p/replay trajectories can also start and end at various track locations other than the track ends, removal of time bins from frames also allowed us to study the impact of swapping of decoded locations beyond the track ends. Removal of up to three-bin epochs from the frames' start or end, but not the frames' middle (as control), reduced the difference between the cumulative proportion of significant frames across groups in the sleep sessions (assessed via two-sided Z-tests for 2 proportions on pooled data across all animals), specifically in Post-Day1Run1 sleep (Fig. 3g, bottom-left, Supplementary Fig. 4c).

Second, by studying short-timescale decoded spatio-temporal relationships within sleep frames, we found that cuboid-reared rats had a robust likelihood to depict similar/nearby locations in consecutive 20-ms time bins in both sleep sessions at all locations across the track. This indicates that averaged sleep ensemble activity in frames consistently depicts spatial experiences. In contrast, sphere-reared rats exhibited an impairment in the specificity (index) of depiction of locations at/near the two track ends (Fig. 3g, right). Meanwhile, the specificity index of distinguishing the track middle from the track ends was not different between groups, suggesting a specific impairment in distinctly representing the two track ends in the sphere-reared rats (Supplementary Fig. 4d). Sphere-reared, but not cuboid-reared rats, had a significant correlation between the proportion of significant sleep p/replay frames (Fig. 2c) and the sleep specificity index for the decoded track ends (cuboid: R = −0.01, p = 0.96; sphere: R = 0.49, p = 0.030, Spearman's Rank-order correlations). Both groups showed no relationship between the proportion of significant sleep p/replay frames and place cells' spatial information during run (cuboid: R = −0.04, p = 0.87; sphere: R = 0.28, p = 0.23, Spearman's rank-order correlations). Track-end specificity was also lower during awake-rest frames in sphere-reared rats (Supplementary Fig. 4e), indicating that, while the proportions of awake-rest replay were similar across groups (Fig. 2d), there remained a deficit in track-end depiction.

Third, decoding trajectories during sleep frames after discarding place maps near the track ends, specifically to study their impact on the difference in decoded trajectory across groups, attenuated the differences in proportion of significant decoded p/replay frames between cuboid- and sphere-reared rats (Supplementary Fig. 4f). A similar truncation analysis for the track middle (as control) had no significant impact on the difference across groups, which demonstrates a preferential effect of this spatial deprivation on neural activity at track ends (Supplementary Fig. 4g). Finally, use of circular statistics to quantify p/replay trajectory sequences depicting the linear track attenuated the observed differences in trajectory sequences between groups, likely contributed by an increased tendency for circularity in the sphere-reared group, specifically in replay (Supplementary Fig. 4h). Overall, these multiple lines of evidence suggest a degree of representational 'warping' of linear space (i.e., a deviation from linearity, during run and sleep, more apparent in trajectory replay compared to preplay) in the sphere-reared rats during development.

### Relative reduction in intrinsic repertoire of clustered neuronal ensembles during sleep after sphere-rearing

Given the increased likelihood of trajectory orthogonalization and segmentation in cuboidal vs. spherical environments due to presence of 90-degree corners and linear vertical borders (Fig. 1a), we wondered if these different experiences across development shaped animals' internal models of space accordingly. Thus, we investigated the structure of spontaneous activity of hippocampal neuronal ensembles during Pre-Day1Run1 sleep, while inputs from the external world were minimally processed. Activity of pairs of putative pyramidal cells

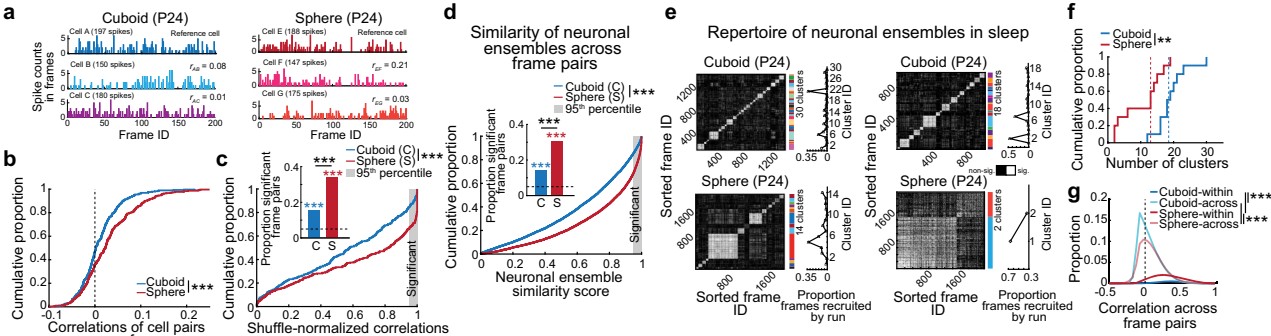

**Fig. 4 | Reduced repertoire of distinct neuronal ensembles during sleep after sphere-rearing. a** Examples of spike count correlations between cell pairs across 200 frames for 3 pyramidal cells during Pre-Day1Run1 sleep in cuboid- (left) and sphere-reared (right) rats. Top: reference cell for correlations. Total spike counts are given in brackets next to cell ID. **b** Cumulative distributions of correlations of spike counts between pyramidal cell pairs across sleep frames. Note higher values in sphere- vs. cuboid-reared rats ($p = 2.7 \times 10^{-5}$, two-sided rank-sum test). **c** Shuffle-normalized correlations of cell pairs are higher in sphere- vs. cuboid-reared rats ($p = 7.3 \times 10^{-6}$, two-sided rank-sum test). Inset: proportion of significantly correlated cell pairs (>95th percentile of shuffles) is higher in sphere- vs. cuboid-reared rats ($p = 1.9 \times 10^{-7}$, two-sided Z-test for 2 proportions). **d** Similarity scores ($p = 3.0 \times 10^{-301}$, two-sided rank-sum test) and proportions of correlated Pre-

Day1Run1 sleep frame pairs (inset; $p = 0$, one-sided Binomial tests vs. chance for both groups) vs. chance. Note higher scores and proportions in sphere-reared vs. control rats ($p = 1.9 \times 10^{-238}$, two-sided Z-test for 2 proportions). **e** Examples of clustered Pre-Day1Run1 sleep frames with correlated activity in cuboid- (top) and sphere-reared (bottom) rats. Colorbar: color-coded clusters and cluster demarcation. Curves to the right of cluster panels: Proportions of frames from each cluster that are significantly correlated with the future run trajectory. **f** Number of clusters during Day1 sleep sessions (cuboid > sphere, $p = 0.0051$, two-sided rank-sum test). $N = 10$ (5 rats/group; 2 sleep sessions/rat). **g** Distributions of Pre-Day1Run1 sleep frame-pair correlations (within>across clusters, $p = 0$; sphere vs. cuboid, within or across clusters: $p = 0$, two-sided exact permutation tests). **b–d**, **g** $N = 5$ rats/group. ***$p < 0.005$. **$p < 0.01$. Source data are provided as a Source Data file.

during Pre-Day1Run1 sleep (defined for each neuron as their spike counts in each of all the sleep frames) exhibited significantly higher correlations in the sphere-reared compared with cuboid-reared rats (Fig. 4a, b). These remained significant after normalizing them by firing rate-matched shuffled correlations (expressed from 0 to 1), to account for any contributions of firing rates[35] (Fig. 4c). A higher proportion of cell pairs were significantly correlated in the sphere- vs. cuboid-reared rats (34% vs. 15% at a cutoff p-value of 0.05, Fig. 4c inset). The place cell-pair correlations across the sleep frames in the naïve sphere-reared rats were unrelated to the primary place field lengths on the linear track during Day1Run1 (Pearson's correlation R = 0.037, $p = 0.36$), and remained higher in down-sampled datasets with matched spatial information or primary place field length across groups (Supplementary Fig. 5a).

We next investigated whether the increase in sleep cell-pair correlations after sphere-rearing (Fig. 4a–c) translated into an increased similarity in neuronal ensembles activated across frames, which estimates the richness of the sleep ensemble activity. For each sleep frame we generated a population vector composed of the spike counts (≥0) emitted in the frame by the recorded putative pyramidal neurons. For each frame pair, we derived a neuronal ensemble similarity score (i.e., the shuffle-normalized correlation ranging from 0 to 1) between the corresponding pair of neuronal population vectors. In agreement with our cell-pair results (Fig. 4a–c), we found that neuronal ensembles activated across pairs of sleep frames were more similar in naïve sphere-reared compared with cuboid-reared rats (Fig. 4d). Groups of sleep frames sharing similar neuronal ensembles formed distinct clusters (detected using k-means clustering algorithms[36]; Fig. 4e, Supplementary Fig. 5b). Neuronal ensembles active within a subset of sleep clusters were later preferentially recruited to represent future Day1Run1 experiences, indicating these preconfigured ensembles are utilized for representation of future novel experiences[1] (Fig. 4e, right). Importantly, cuboid-reared rats had a larger repertoire of clusters compared to sphere-reared rats (cuboid vs. sphere median number of clusters: 18.5 vs. 13 in Day1 sleep sessions, $p = 0.0051$ for Day1 sleep sessions, $p = 0.048$ for Pre-Day1Run1 sleep sessions, two-sided rank-sum tests, Fig. 4f, Supplementary Fig. 5c–g, which differed from shuffles in both groups). The within-cluster correlations were higher than the across-cluster ones (within>across clusters in both groups,

$p = 0$; sphere vs. cuboid, within clusters or across clusters: $p = 0$, exact-permutation tests, Fig. 4g). No further increase in the cluster repertoire size was observed in a control group of cuboid-reared naïve adult rats ($N = 3$; Supplementary Fig. 5h, i), indicating that at P23-24 it has acquired adult-like features. In addition to fewer clusters, sphere-reared rats had lower intra-cluster and higher between-cluster similarities compared with cuboid-reared rats (Fig. 4g). Furthermore, geometric experience with linearity during Day1Run1 reduced the similarity of activated neuronal ensembles across Post-Day1Run1 sleep frames in sphere-reared rats (Supplementary Fig. 5j). We propose that deprivation from experience with geometric linearity (replaced with curvature) by sphere-rearing reduced the intrinsic cluster repertoire size.

### Sphere-rearing impaired neuronal pattern separation and plasticity of multiple linear experiences

As the internal model of space of sphere-reared rats expressed during sleep relied on a relatively reduced repertoire of orthogonal neuronal ensembles (Fig. 4), we hypothesized this could reduce the network ability to simultaneously represent multiple distinct linear tracks. Given that sphere-reared rats were deprived of experience with right-angle corners (Fig. 1a), we exposed our rats to alternating experiences on three distinct 1 m-long linear tracks arranged in U-shape, separated by 90° corners[10]. On Day1, rats were familiarized with running on track 1 (Figs. 1 to 4). On Day2, rats slept in a corresponding-geometry box (Pre-Day2Runs sleep), then explored track 1 in isolation followed by exploration of the contiguous 3 tracks (tracks 1–3, Day2Run2), slept again in the box (Post-Day2Runs sleep), and re-explored the 3 tracks (Fig. 5a).

Since it remained unknown when the hippocampal network first acquires the ability to distinctly represent, or pattern separate, multiple linear spatial experiences[37–39] during early postnatal life, we first studied pattern separation of 3 distinct linear tracks in younger, P16-25 cuboid-reared rats ($N = 19$ rats undergoing the Day2 experimental protocol for the first time at various developmental ages, Methods). We started from P16, when time-compressed sequences are not yet expressed, and continued until P25, when preplay, theta sequences, and plasticity in replay have emerged[11]. Neuronal ensembles failed to discriminate the 3 tracks at younger ages. Pattern

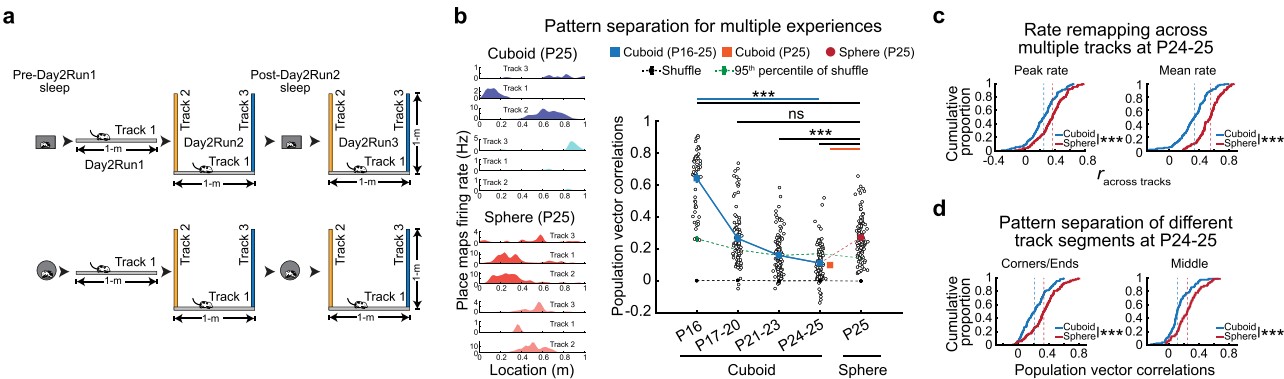

**Fig. 5 | Impaired pattern separation of distinct multiple linear track experiences after sphere-rearing. a** Experimental design (Day2): sleep and run on 3 linear tracks. **b** Developmental emergence of place cell re/mapping and pattern separation across 3 tracks. Left: examples of re/mapping in cuboid- and sphere-reared rats. Right: age-related increase in pattern-separation (re/mapping) in P16-P25 cuboid-reared rats ($p = 2.3 \times 10^{-72}$, ANOVA) and its impairment in P25 sphere-reared rats (P25 sphere vs. P16 cuboid: $p = 7.2 \times 10^{-19}$; P25 sphere vs. P17-20 cuboid: $p = 0.23$; P25 sphere vs. P21-23 cuboid: $p = 2.1 \times 10^{-10}$, P25 sphere vs. P24-25 cuboid: $p = 4.6 \times 10^{-17}$, P25 sphere vs. P25 cuboid: $p = 1.8 \times 10^{-16}$, two-sided rank-sum tests). Data are means ± SEM. N (rats/group x 2 directions/rat for each track): Cuboid: 6 (P16), 10 (P17-20), 12 (P21-23), 10 (P24-25), 8 (P25); Sphere: 10 (P25). **c** Reduced discriminability between multiple tracks (i.e., increased peak and mean firing-rate correlations) after sphere-rearing (P24-25 cuboid-reared vs. P25 sphere-reared: $p = 7.8 \times 10^{-5}$ and $p = 1.8 \times 10^{-13}$ respectively, two-sided rank-sum tests). **d** Higher similarity between mapping of different corners/ends ($p = 8.1 \times 10^{-6}$, two-sided rank-sum test, left) and tracks' middle across the 3 tracks ($p = 6.9 \times 10^{-5}$, two-sided rank-sum test, right) in the sphere-vs. cuboid-reared rats. N (rats/group x 2 directions/rat for each track): (**c, d**) Cuboid, P24-25 (10); Sphere, P25 (10). ***$p < 0.005$. ns = not significant. Source data are provided as a Source Data file. Cartoons in Fig. 5/Panel a adapted from U. Farooq, G. Dragoi, Emergence of preconfigured and plastic time-compressed sequences in early postnatal development. Science 363, 168–173 (2019). DOI: 10.1126/science.aav0502. Reprinted with permission from AAAS.

separation emerged and improved with age, as shown by a marked age-dependent reduction in correlations between the ensembles recruited across pairs of the 3 experiences (Fig. 5b), likely due to developmental hippocampal maturation[11,40]. Interestingly, P25 sphere-reared rats had impaired pattern separation compared to P21-23 and P24-25 cuboid-reared rats (Fig. 5b), likely contributed by higher correlations between population firing rate-representations of the 3 tracks (Fig. 5c). The deficit was observed for pairs of novel tracks and was not due to differences in directional coding (Supplementary Fig. 6a–c). The increased similarity for multiple tracks in the sphere-reared rats was observed for mapping right-angle corners separating orthogonal track-pairs and track ends, and extended to other track locations (Fig. 5d).

Experience with multiple contexts over short time intervals, as in our Day2 experiment, could challenge the hippocampal network to distinctly encode and store these experiences due to possible memory interference[41]. A potential solution for this problem would be the availability and selection of specific, unique preconfigured preplay sequences during encoding of distinct run experiences, thus reducing the interference in their representation and subsequent long-term storage[1,10]. Both sphere- and cuboid-reared P24-25 animals exhibited significant preplay of the 2 novel tracks during the Pre-Day2Run2 sleep (Fig. 6a, cuboid, sphere: $p = 4 \times 10^{-5}$, $p = 0.0026$, paired one-sided $t$-tests vs. shuffle, yellow stars; $p = 1.7 \times 10^{-6}$, $p = 3 \times 10^{-4}$, one-sided $t$-tests vs. $p = 0.05$ chance, black stars; $p = 0$, $p = 0$, one-sided Binomial tests vs. $p = 0.05$ chance, respectively), like adult rats[10]. Similar to Pre-Day1Run1 (Fig. 2c), sphere-reared rats exhibited lower incidence of preplay compared to their cuboid-reared counterparts on Pre-Day2Run2 sleep (Fig. 6a, $p = 6.3 \times 10^{-4}$, two-sided $t$-test, Supplementary Fig. 6d for supporting preplay analyses). Next, we studied if preplay uniquely depicted each of the multiple tracks by assessing the incidence of track-specific preplay (i.e., frames which selectively depicted sequential trajectories for only one novel, but not the other two tracks). Track-specific preplay occurred above chance levels in the cuboid-reared rats (Fig. 6a), while in the sphere-reared rats, the incidence of track-specific preplay was lower ($p = 0.0013$, two-sided $t$-test) and did not exceed chance levels (Fig. 6a, cuboid, sphere: $p = 0.0015$, $p = 0.50$,

paired one-sided $t$-tests vs. shuffle, yellow stars; $p = 8.3 \times 10^{-4}$, $p = 0.88$, one-sided $t$-test vs. $p = 0.05$ chance, black stars; $p = 3.7 \times 10^{-7}$, $p = 0.93$, one-sided Binomial tests vs. $p = 0.05$ chance, respectively). This indicates a deficit in generating distinct temporal codes for multiple future experiences during sleep in the sphere-reared rats.

Reduced track-specific sleep preplay and pattern separation during run in the sphere-reared rats could, in part, be due to a higher similarity between neuronal ensembles across prior sleep frame pairs, as on Day1 (Fig. 4d–f). Indeed, the proportions of significantly correlated frame pairs during Pre-Day2Run2 sleep remained higher after sphere-rearing (Fig. 6b). While this difference did not directly translate into statistically significant lower numbers of clusters of neuronal ensembles (Fig. 6b), they were associated with less-defined cluster boundaries in the sphere-reared rats (Supplementary Fig. 6e). Thus, while these results indicate that prior Day1 experience with geometric linearity (Supplementary Fig. 5h) and subsequent overnight sleep partially improved the cluster repertoire in the sphere-reared rats, the repertoire remained less defined on Day2. Clusters of neuronal ensembles in cuboid-reared, but not sphere-reared, rats were recruited during each of the 3 future track experiences, with minimal overlap, indicating they likely contributed to the distinct encoding of multiple tracks in the cuboid-reared rats (Fig. 6b, bottom-right). These deficits in the sphere-reared rats also extended to experienced-trajectory replay (Fig. 6c, cuboid replay, track-specific replay: $p = 9.9 \times 10^{-7}$, $p = 6.1 \times 10^{-4}$, paired one-sided $t$-tests vs. shuffle, yellow stars; $p = 2.7 \times 10^{-8}$, $p = 0.0014$, one-sided $t$-tests vs. $p = 0.05$ chance, black stars; $p = 0$, $p = 1.5 \times 10^{-5}$, one-sided Binomial tests vs. $p = 0.05$ chance; sphere replay, track-specific replay: $p = 0.0079$, $p = 0.86$, paired one-sided $t$-tests vs. shuffle, yellow stars; $p = 9.6 \times 10^{-4}$, $p = 0.95$, one-sided $t$-tests vs. $p = 0.05$ chance, black stars; $p = 0$, $p = 1$, one-sided Binomial tests vs. $p = 0.05$ chance, respectively; cuboid > sphere replay and track-specific replay: $p = 1.1 \times 10^{-5}$, $p = 8.4 \times 10^{-4}$, two-sided $t$-tests; Supplementary Fig. 6f for supporting replay analyses).

To investigate the experience-related plasticity for multiple experiences, we isolated cell-assemblies active within 20 ms time bins that were co-activated during the novel runs and quantified their activation in the sleep sessions flanking the experiences. We

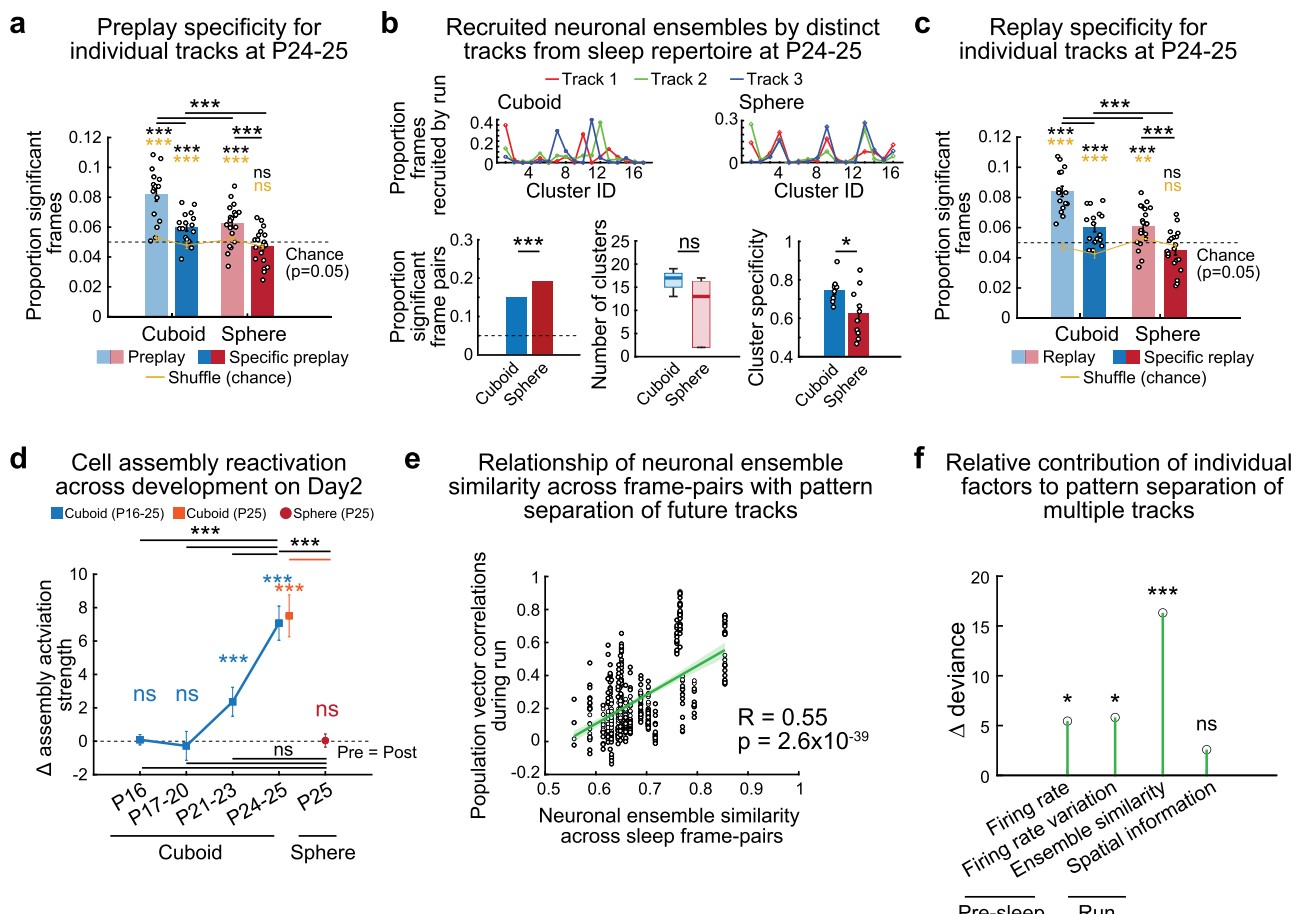

**Fig. 6 | Reduced neuronal ensemble track-specific depiction of future and reactivation of past distinct multiple linear track experiences during sleep after sphere-rearing. a** Proportions of preplay and 'specific-track' preplay after cuboid- and sphere-rearing. Yellow/black stars: *p*-values for statistical comparison with shuffle and chance. **b** Experiences on distinct tracks during Day2Run2 recruit neuronal ensembles from a larger repertoire of Pre-Day2Run1 sleep frame clusters in cuboid- vs. sphere-reared rats (individual rat examples). Proportion of significant frame pairs in Pre-Day2Run1 sleep remains significantly higher in sphere- vs. cuboid-reared rats (bottom-left, $p = 6.0 \times 10^{-16}$, two-sided *Z*-test for 2 proportions), which does not translate to a significant difference in the number of discrete clusters (bottom-center, $p = 0.16$, two-sided rank-sum test). Specificity (bottom-right) of cluster recruitment from Pre-Day2Run1 sleep by each future track is higher in cuboid- vs. sphere-reared rats ($p = 0.030$, *t*-test). **c** Proportions of replay and 'specific-track' replay after cuboid- and sphere-rearing. Yellow/black stars: *p*-values for statistical comparison with shuffle and chance. **d** Cell assembly plasticity (sleep reactivation exceeding pre-activation) for multiple experiences in P16-25 cuboid-reared and P25 sphere-reared rats. Cuboid-reared P24-25 rats exhibit multi-

fold increase in experience-dependent cell assembly plasticity (sleep reactivation) vs. younger rats. Cell-assemblies in sphere-reared rats (P25) have impaired plasticity compared with P24-25 and age-matched (P25) cuboid-reared rats, and similar reactivation to P16-23 cuboid-reared rats. **e** Average neuronal ensemble similarity across frame-pairs in Pre-Day2Run1 sleep is significantly correlated with average population vector correlations across future track pairs during run (Pearson's correlation, *N* = 24 rats). Shaded region is 95% confidence interval. **f** Contribution of Pre-Day2Run1 sleep or run firing properties to pattern separation of multiple spatial experiences estimated using multiple regression (*N* = 24 rats). Sleep individual neuronal firing rates ($p = 0.020$), across-neuron firing rate variation ($p = 0.016$), neuronal ensemble similarity across frame-pairs ($p = 5.4 \times 10^{-5}$) are all significantly related to population vector correlations across future distinct track pairs, while run spatial information has no further significant relationship ($p = 0.11$, likelihood ratio tests). Data in a–d are represented as mean ± SEM. *N* (rats/group x 2 directions/rat for each track): (**a**–**c**) Cuboid, P24-25 (8); Sphere, P25 (10).
***$p < 0.005$. **$p < 0.01$. *$p < 0.05$. ns = not significant. Source data are provided as a Source Data file.

first studied the neurodevelopment of cell assembly plasticity for multiple, novel experiences (Fig. 6d). Young cuboid-reared rats (P16-20) lacked cell assembly plasticity for multiple experiences, in agreement with prior findings on single context experiences[11,42] ($p = 0.97$, $p = 0.85$ for P16, P17-20, two-sided signed-rank tests). While P21-23 cuboid-reared rats exhibited limited significant plasticity, P24-25 cuboid-reared rats had markedly higher plasticity than younger rats, suggesting that as pattern separation improved with age, so did the ability to store these experiences ($p = 0.00039$ for P21-23, $p = 8.9 \times 10^{-5}$ for P24-25, and $p = 0.00044$ for P25, two-sided signed-rank tests; reactivation in P24-25 is greater than P16, P17-20, P21-23 cuboid-reared rats: $p = 1.1 \times 10^{-5}$, $p = 1.0 \times 10^{-5}$, $p = 0.00020$, two-sided rank-sum tests). Interestingly, the deficits in pattern separation in the P25 sphere-reared rats led to a marked reduction

in the overall experience-induced plasticity[16,29–31] during Post-Day2Runs sleep, similar to younger P16-23 cuboid-reared rats (Fig. 6d; reactivation in P25 sphere-reared rats is lower than P24-25 and age matched P25 cuboid-reared rats $p = 3.9 \times 10^{-7}$ and $p = 2.7 \times 10^{-6}$, rank-sum tests; reactivation in P25 sphere-reared rats is similar to P16, P17-20, P21-23 cuboid-reared rats, $p = 0.50$, $p = 0.75$, $p = 0.052$, two-sided rank-sum tests; N (rats/group x 2 directions/rat for each track): Cuboid: 6 (P16), 10 (P17-20), 10 (P21-23), 8 (P24-25), 6 (P25); Sphere: 10 (P25)). While P25 sphere-reared rats lacked plasticity (reactivation > pre-activation, $p = 0.41$, two-sided signed-rank test), they exhibited a significant geometric experience-dependent reorganization of sleep cell assembly dynamics suggesting the experience reorganizes the network (Supplementary Fig. 6g). We propose the increased similarity of neuronal ensembles'

activity during sleep after sphere-rearing prevented the expression of unique preplay for distinct future run experiences, which impaired both pattern separation during track(s) exploration and sleep reactivation.

## Sleep repertoire is linked with pattern separation of multiple future spatial experiences

We further probed the relationship between the intrinsic sleep network dynamics and the ability to generate distinct neuronal ensembles to discriminate multiple future experiences (Fig. 5b). Pre-Day2Run1 sleep neuronal ensemble similarity across frame-pairs (a metric of sleep repertoire richness, where higher similarity denotes lower richness) was significantly correlated with the place cell population vector correlations across different future linear track experiences (Fig. 6e; Pearson's correlation, $R = 0.55$, $p = 2.6 \times 10^{-39}$, for track-pairs from $N = 24$ rats: 19 cuboid-reared rats across development and 5 sphere-reared rats).

To assess the contribution of various factors to pattern separation, we constructed a multiple regression model, with population vector correlation across tracks as the dependent variable, and rat age, rearing environment, various single-cell and ensemble properties of the Pre-Day2Run1 sleep (unrelated to run for unbiased evaluation), and run as predictors (Methods). We confirmed that both rat age (P16-25, β-coefficient: $-0.036 \pm 0.004$, $p = 3.8 \times 10^{-20}$, t-test) and rearing environment (cuboid or sphere, β-coefficient: $0.159 + 0.023$, $p = 9.1 \times 10^{-12}$, t-test) were significant predictors. In addition, individual neuronal firing rates during Pre-Day2Run1 sleep (β-coefficient: $0.034 \pm 0.015$, $p = 0.021$, t-test), Pre-Day2Run1 sleep firing rate variation across neurons (β-coefficient: $-0.073 \pm 0.030$, $p = 0.017$, t-test), and neuronal ensemble similarity across frame pairs (β-coefficient: $1.609 \pm 0.398$, $p = 6.2 \times 10^{-5}$, t-test) were all predictors of population vector correlations across tracks (Fig. 6f). Meanwhile, place cell spatial information during run was not a significant predictor (β-coefficient: $-0.030 \pm 0.019$, $p = 0.11$, t-test). Thus, individual-cell firing rate, across-neuron firing rate variation, and the ensemble sleep repertoire, all during sleep, affect pattern separation of future tracks (Fig. 6f), indicating that network preconfiguration constrains hippocampal representation of future distinct spatial experiences at multiple levels.

## Extended experience with geometric linearity after sphere-rearing reconfigures neuronal ensembles during run

We asked if the deficits induced by early-life deprivation from geometric linearity could be countered following experiences on multiple linear tracks across days. Thus, on Days3–4 we exposed our rats to alternating sessions of sleep and run on multiple linear tracks (Fig. 7a) and compared the changes in neuronal representations across days and groups. The tracks explored on Day3 were either previously experienced over multiple days (track 1, familiar) or were novel (tracks 4–5), allowing us to study the influence of experience with geometric linearity on representations of familiar tracks and in latently shaping dynamics for representing novel tracks. Similarly, on Day4, we re-exposed our rats to a familiar track (track 4) followed by exposure to a novel track (track 6).

Experience over days with multiple linear tracks markedly improved spatial tuning (Figs. 1, 3, 5). While primary place field length was reduced to a greater degree in sphere-reared rats on novel tracks on Day3 compared with Day1, it became similar across groups on Day4 (Fig. 7b). Consistently, spatial information on novel tracks became similar across groups on Day4, exhibiting an accelerated recovery for familiar tracks on Day3 (Supplementary Fig. 7a–c).

Ensemble representation of opposite track-ends was more distinct in the sphere-reared rats on Day3 than Day1 and became similar with that of cuboid-reared rats during Day4Runs (Fig. 7c–d). Similarly, the altered representation of novel linear space in sphere-reared rats on Day1 (Fig. 3d) recovered on Day3 for novel tracks,

together with a similarly low proportion of track-symmetric and low-tuned individual rate maps across groups (Supplementary Fig. 7d–e). Repeated experience over days on the same track (Days1-3, track 1) led to an accelerated recovery in the representation of familiar linear space (Supplementary Fig. 7f–h). Finally, we investigated the effect of extended experience with geometric linearity on the ability to pattern separate between distinct linear tracks. We observed an improvement in pattern separation for multiple linear experiences on Day3 compared with Day2 specifically in the sphere-reared rats (Fig. 7e), which became similar across groups by Day4 (Fig. 7e, bottom-right).

## Extended experience with geometric linearity after sphere-rearing restores the richer intrinsic sleep repertoire recruited by future experiences

We next determined if experience with geometric linearity across days could reconfigure the intrinsic sleep network activity patterns as well as their selection by future experiences (Figs. 2, 4, 6). As on Days 1–2, sphere-reared rats had significant p/replay of future run trajectories during the Pre-Day3Runs sleep, which remained lower in sphere-reared rats (all tracks explored within a run session following sleep, Fig. 8a, left, cuboid vs. shuffle: $p = 6.0 \times 10^{-8}$, sphere vs. shuffle: $p = 3.1 \times 10^{-8}$, paired one-sided t-tests; $p = 3.5 \times 10^{-9}$, $p = 6.2 \times 10^{-10}$ one-sided t-test vs. $p = 0.05$ chance for the two groups; $p = 0$, one-sided Binomial tests vs. $p = 0.05$ chance for both groups; cuboid > sphere: $p = 0.045$, two-sided t-test). On Day4, p/replay became similar across the two groups, indicating sphere-reared rats showed a gradual experience dependent improvement in p/replay sequences (Fig. 8a, right; cuboid vs. shuffle: $p = 1.5 \times 10^{-9}$, sphere vs. shuffle: $p = 1.8 \times 10^{-5}$, paired one-sided t-tests; cuboid: $p = 5.8 \times 10^{-11}$, sphere: $p = 5.1 \times 10^{-6}$, one-sided t-tests vs. $p = 0.05$ chance; $p = 0$, one-sided Binomial tests vs. $p = 0.05$ chance for both groups; cuboid vs. sphere p/replay, $p = 0.12$, two-sided t-test). Additionally, future track-specific p/replay first emerged on Day3 in the sphere-reared rats (Fig. 8a, left, cuboid vs. shuffle: $p = 2.9 \times 10^{-5}$, sphere vs. shuffle: $p = 0.0012$, paired one-sided t-tests; cuboid: $p = 1.8 \times 10^{-4}$, sphere: $p = 0.018$, one-sided t-tests vs. $p = 0.05$ chance; $p = 2.2 \times 10^{-16}$ and $p = 1.4 \times 10^{-6}$, one-sided Binomial tests vs. $p = 0.05$ chance for each group). Future track-specific p/replay became similar to cuboid-reared rats by Day4 (Fig. 8a, right, cuboid vs. shuffle: $p = 1.3 \times 10^{-6}$, sphere vs. shuffle: $p = 0.0041$, paired one-sided t-tests; cuboid: $p = 1.5 \times 10^{-6}$, sphere: $p = 0.0054$, one-sided t-tests vs. $p = 0.05$ chance; $p = 0$ and $p = 1.7 \times 10^{-6}$, one-sided Binomial tests vs. $p = 0.05$ chance for each group; cuboid vs. sphere p/replay: $p = 0.32$, two-sided t-test; Supplementary Fig. 8a for supporting analyses; Supplementary Fig. 8b–d for novel tracks only on Day4). Consistently, cell assemblies expressed during novel runs on Days3–4 exhibited similar plasticity across groups (Supplementary Fig. 8e).

We next evaluated the influence of geometric experience with linearity on the intrinsic sleep dynamics (Figs. 4, 6b). Similarity of pyramidal cell-pair activity across sleep frames (Fig. 8b, left) and the similarity of ensemble frame-pair activity during Day4 reduced compared to Day1 in the sphere-reared rats (Fig. 8b, right). Experience with geometric linearity enriched the intrinsic sleep repertoire of distinct neuronal ensembles, increasing the number of distinct clusters, whose selection during run contributed to a better discrimination of the multiple, distinct linear tracks in the sphere-reared rats (Fig. 8c–d).

Altogether, repeated experience with geometric linearity over four days, starting from the de novo learning of a linear space, largely countered the multiple effects due to sphere-rearing from birth, as expressed during sleep, run, and in the relationship of sleep with run (Figs. 7, 8, Supplementary Figs. 7, 8; Fig. 8e models the concerted recovery of 6 major neuronal parameters). These improvements might have been facilitated by our extensive protocol, which involved repeated training on up to 6 distinct linear contexts with different geometric relationships.

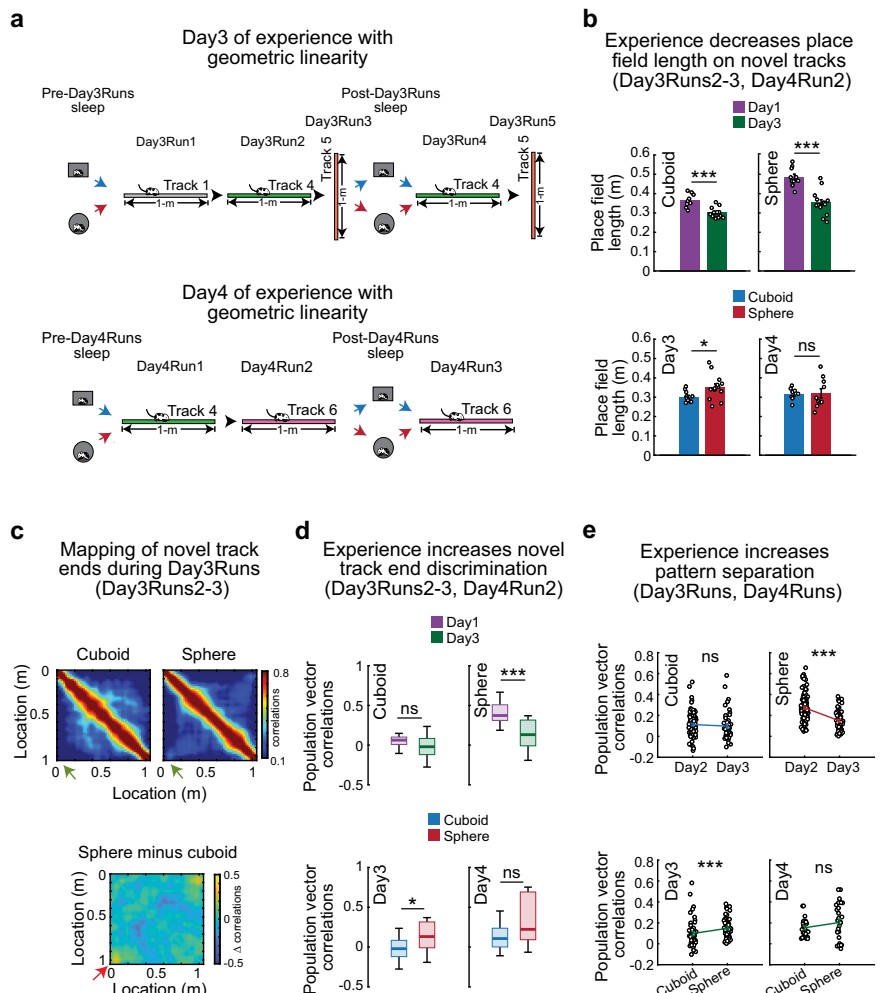

**Fig. 7 | Repeated experience with geometric linearity across four days largely rescues the deficits observed during run associated with sphere-rearing from birth. a** Experimental design on Day3 (top) and Day4 (bottom): alternating sleeps and runs on multiple linear tracks. **b** Experience-dependent changes in primary place field length on novel tracks across experimental days. Higher reduction in place field length across days in sphere- (top-right, $p = 0.00025$) vs. cuboid-reared rats (top-left, $p = 0.00068$). Significant residual difference between groups on Day3 (bottom-left, $p = 0.025$), but not on Day4 (bottom-right, $p = 0.76$, two-sided rank-sum tests). **c** Correlation matrix of neuronal ensemble mapping of locations along the linear track during Day3Runs2-3 for cuboid- (top-left), sphere-reared rats (top-right), and the difference between groups (bottom). Green/red arrows mark increased residual correlations on Day3 at track ends in sphere-reared rats. **d** Population vector correlation of place maps for track ends on novel tracks on Day3 compared with Day1. Note improved discrimination across days in sphere- (top-right, $p = 0.0031$) but not cuboid-reared rats (top-left, $p = 0.28$) and the

residual difference between groups on Day3 (bottom-left, $p = 0.025$). Track end discrimination for novel tracks became similar for cuboid- and sphere-reared rats on Day 4 (bottom-right, $p = 0.24$, two-sided rank-sum tests). **e** Increased pattern-separation of multiple tracks on Day3 over Day2 (top-right) in sphere- ($p = 1.5 \times 10^{-8}$) but not cuboid-reared rats (top-left, $p = 0.12$, two-sided rank-sum tests, left). Day3 pattern separation across groups: $p = 0.0016$, two-sided rank-sum test (bottom-left). Pattern separation of place cell maps for multiple tracks became similar across groups on Day4 (bottom-right, $p = 0.28$, two-sided rank-sum test). Data in **b**, **e** are represented as mean ± SEM. $N$ (rats/group x 2 directions/rat for each track) on Day3 and 4: Cuboid-reared (P25-26 and P26-27, respectively)=6; Sphere-reared (P26 and P27, respectively)=8 (detailed in Methods). ***$p < 0.005$. *$p < 0.05$. ns = not significant. Source data are provided as a Source Data file. Cartoons in Fig. 7/Panel a adapted from U. Farooq, G. Dragoi, Emergence of preconfigured and plastic time-compressed sequences in early postnatal development. Science 363, 168–173 (2019). DOI: 10.1126/science.aav0502. Reprinted with permission from AAAS.

## Discussion

We have demonstrated that depriving rats of experience with crucial features of Euclidean geometry (Fig. 1a), which are likely individually encountered in the external world, via sphere-rearing across development affected individual place cell tuning for linear space. Yet sphere-rearing did not block the emergence of preconfigured and plastic hippocampal time-compressed neuronal sequences that encode and consolidate novel experiences on a linear track. Instead, early-life experience with linearity, vertical boundaries, planarity, and corners was critical for the emergence of an enriched repertoire of preconfigured neuronal sequences, enabling discrimination of distinct spatial features and experiences on multiple linear tracks. Repeated experience with geometric linearity over 4 days significantly reshaped neuronal ensembles, enriching their repertoire, and improved pattern

separation of multiple experiences. Thus, while dispensable for the representation of many features of a generic linear track[43], prior experience with Euclidean geometry proved more critical for the distinct representation of multiple linear tracks and track ends and corners via rapid pattern separation during early development. We posit that a change in the nature of experience across the two rearing environments was associated with a change in the repertoire of hippocampal network motifs. Whether the default repertoire is the one observed after sphere-rearing, which is enriched via experience with Euclidean geometry or, conversely, the default repertoire is the one observed after cuboid-rearing, which is reduced by un-natural deprivation via sphere-rearing, remains to be determined. Regardless, inferred intrinsic developmental programs and explicit geometric experience might play different and not completely overlapping roles

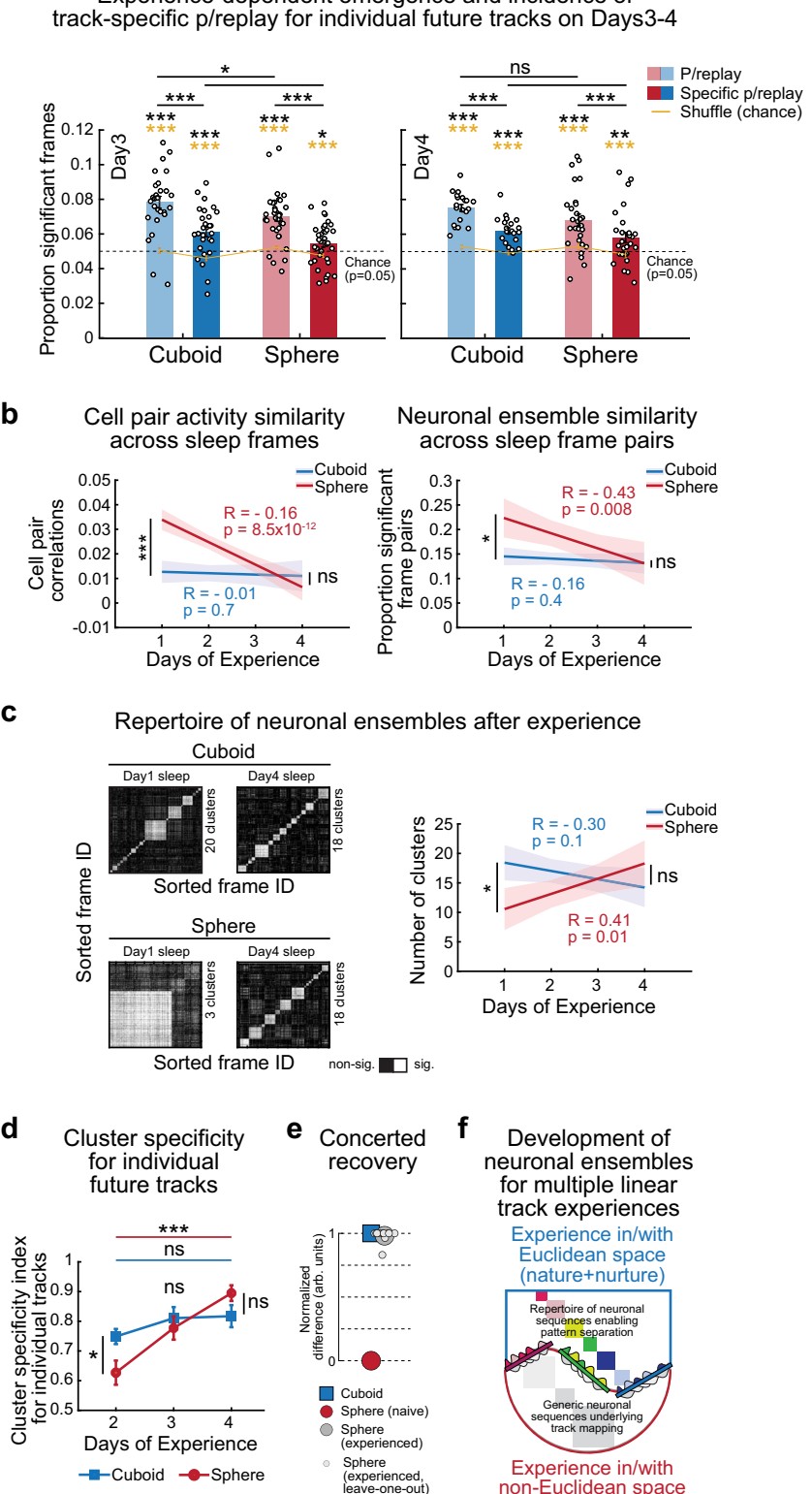

**a** Experience-dependent emergence and incidence of track-specific p/replay for individual future tracks on Days3-4

**b** Cell pair activity similarity across sleep frames

Neuronal ensemble similarity across sleep frame pairs

**c** Repertoire of neuronal ensembles after experience

**d** Cluster specificity for individual future tracks

**e** Concerted recovery

**f** Development of neuronal ensembles for multiple linear track experiences

in the expression of more complex repertoires in the cuboid-reared group. From a nature vs. nurture debate perspective, early-life experience with features of Euclidean geometry (i.e., nurture) increases hippocampal network 'performance' to rapidly discriminate between multiple linear environments[24], while network 'competence' to express a preconfigured, generic representation of linear environments[24] develops a priori (i.e., nature) (Fig. 8f).

Earlier studies did not discover a critical role of early-life sensory experience in the development of spatial circuits in the brain, including place cells and head-direction cells[44,45]. Recent evidence indicates that geometric experience, and not simply sensory, including visual, experience, mildly influences the expression of individual grid cells in the adult medial entorhinal cortex during exploration of an individual spatial context[46]. This is in contrast with the marked effects of early-life

**Fig. 8 | Repeated experience with geometric linearity across four days enriches the intrinsic sleep repertoire in sphere-reared rats. a** Incidence of p/replay and track-specific-exclusive p/replay for future tracks were above chance across groups and higher in cuboid-reared rats on Day 3 (left), and above chance and similar between groups on Day 4 (right). Yellow/black stars: p-values for statistical comparison with shuffle/chance. **b** Experience-dependent reduction in proportion of pyramidal cell-pair correlations across sleep frames in sphere-reared rats (left, cuboid-reared vs. sphere-reared Pearson's correlations; Day1, $p = 6.5 \times 10^{-11}$; Day4, $p = 0.54$, two-sided rank-sum tests) and in proportion of significantly correlated neuronal ensembles across sleep frames in sphere-reared rats (right, cuboid-reared vs. sphere-reared Pearson's correlations; Day1, $p = 0.025$; Day4, $p = 0.14$, two-sided t-tests). **c** Experience on linear tracks increased the number of sleep-frame clusters exclusively in the sphere-reared rats (left, representative examples from same rats across days; right: cuboid-reared vs. sphere-reared Pearson's correlations; Day1, $p = 0.0051$; Day4, $p = 0.37$, two-sided rank-sum tests). **d** Significant increase in cluster specificity for multiple individual tracks from Pre-Day2Run1 sleep to Pre-Day3Runs ($p = 0.019$) and Pre-Day4Runs ($p = 8.6 \times 10^{-5}$) sleep sessions in sphere-reared but not cuboid-reared rats ($p = 0.18$, $p = 0.14$, two-sided t-tests), resulting in similar cluster specificity in sphere- vs. cuboid-reared rats on Pre-Day4Runs sleep sessions ($p = 0.10$, two-sided t-test). **e** Concerted recovery of six individual neuronal parameters (Methods) after 4 days of experience with geometric linearity in sphere-reared rats. Quantification of recovery: proportion between-group difference in parameter values on Day4 (large gray dot) along normalized arbitrary scale bounded by the corresponding values on Day1-2 in the sphere- ('zero'; red dot) and cuboid-reared ('one'; blue square) rats. Small gray dots: contribution of each factor to the group average (large dot). **f** Role of early-life geometric experience in development of the hippocampal neuronal ensemble repertoire. Colored/gray squares: frame-pair clusters. Midline: 3 separate linear tracks with place cells recruited from color-matching clusters. In a, d data are mean ± SEM. In **b**, **c** shaded regions are 95% confidence intervals. N (rats/group x 2 directions/rat per track) on Day3 and 4: Cuboid-reared (P25-26 and P26-27, respectively)=6; Sphere-reared (P26 and P27, respectively)=8 (detailed in Methods). ***$p < 0.005$. **$p < 0.01$. *$p < 0.05$. ns = not significant. Source data provided as Source Data file.

geometric experience we observed here in the hippocampus of P25 rats during navigation of multiple spatial contexts in the form of altered pattern separation between contexts and its relationship to the richness of intrinsic network dynamics in the preceding sleep. Since ontogenetically grid cells mature later than hippocampal place cells[47] and neuronal sequences critical for memory formation in the hippocampus[11,14,47], it is likely that the observed deficits and subsequent recovery of individual and ensemble neuronal activity patterns observed here at P26-27 in sphere-reared rats are not due to changes in entorhinal grid cell signals. In agreement, place cell activity is maintained in adult animals in the absence of the direct input from medial entorhinal cortex to hippocampus[48] or specifically to CA1 area[49]. Future work addressing how the entorhinal grid cell ensemble network behaves during active navigation of multiple spatial contexts and during sleep at compressed timescales after altered early-life geometric experience would elucidate its interaction with and response to hippocampal activity.

Our findings provided three lines of evidence for the role that early-life physical and visual experience with geometry had in sculpting hippocampal neuronal circuits crucial for spatial representation and memory formation. First, geometric deprivation reduced both the intrinsic repertoire of preconfigured neuronal ensembles and their recruitment as distinct ensembles across multiple future linear tracks, resulting in impaired plasticity for multiple experiences. The relatively reduced neuronal ensemble repertoire in the pre-experience sleep after sphere-rearing likely contributed towards the reduced network performance to discriminate multiple linear environments expressed as deficits in pattern separation. Meanwhile, network competence[24] of sphere-reared rats to represent a sequential trajectory preplay prior to their very first encounter with a linear track developed a priori. This indicates that innate intrinsic developmental programs and/or experience with various aspects of the external world, other than Euclidean geometry, were sufficient to support the developmental emergence of time-compressed sequential patterns of neuronal firing previously assumed to require prior experience with geometric linearity. These findings suggest that crucial aspects of higher order ensemble representations of space-time emerge according to Kantian apriorism[12] rather than the predicaments of British empiricism[50]. How intrinsic developmental programs[13] and experiences with curved space enable the emergence of network (pre)configuration expressed as preplay for future novel trajectories in linear environments[9] remains to be determined.

Second, rats' extended experience with curved space inside spheres likely contributed to the expression of changed (i.e., warped) and over-generalized representations for multiple distinct tracks and spatial features. Meanwhile, linear borders, right angles, and vertical walls present in cuboid cages, but absent in spheres, likely acted as geometric landmarks[51,52] instructing brain networks to distinctly map start(s) and end(s) of multiple individual spatial trajectories[38]. During path integration, locations along rats' trajectory on a linear track are mapped (and realign) as a function of distance to track start or end in the earlier and later trajectory segments, respectively[53]. Thus, a potential mechanism for the representational warping is the similar ensemble representation of tracks' start and end after sphere-rearing, via landmark- and object-vector coding in the hippocampal formation[54-58] together with distance coding (and realignment). Consistently, an increased experience of running belts leads to a reduced dependence on object coding and increased dependence on place coding in the dentate gyrus, a crucial region for pattern separation[58]. However, the development in early life of these various cell types and forms of neural coding remains presently unknown. Additionally, increased developmental experience of space curvature could contribute towards the emergence of hippocampal trajectory-depicting ensembles suitable for representing curved spaces, which could translate into warped representations for linear space. Future CA1 neuronal recordings from sphere-reared rats on unbounded tracks, like running belts, and multiple curved spaces, could elucidate the contribution of each of these factors to the warped representation of linear space.

Finally, extended experience with linear environments over four days largely reversed the effects of sphere-rearing. Interestingly, the effects of this spatial deprivation extended to new, not yet experienced linear contexts, indicating that early-life experience of geometry sculpts the neuronal ensemble rules for representing novel spaces. We hypothesize this rapid shift in sphere-reared rats toward accurate depiction of geometric linearity was facilitated by our very selective deprivation from geometric linearity, which spared animal locomotion and age-dependent network maturation, by the evolutionary and ecological affordances[59] of linear space for rats, and by the presence of preconfigured preplay. While cubic shapes might not occur frequently in the wild habitat of rodents, spatial features of cubic shapes are ubiquitous in Euclidean world where rodents (and humans) live. Therefore, these spatial features could further configure the hippocampal representation of space via their direct experience in addition to the inferred innate concepts of space. Given the role of hippocampus in representing conceptual spaces[60] and sequences[10] in adult rodents, our findings on the impact of experience of geometric linearity on hippocampal neurodevelopment might extend to representation of broader conceptual domains later in life. Whether the development of a preconfigured adult hippocampal network repertoire for multiple spatial experiences is influenced by experience specifically within a critical period earlier in life, similar to sensory brain networks[61], and which spatial features of Euclidean geometry are critical to that development remain to be determined.

## Methods

### Animal subjects and groups

Long-Evans pregnant female rats at embryonic day 9-10 (E-9-10) or 15-16 (E15-16) or dams with postnatal day 4 (P4) litters from Charles River Laboratories were housed in a temperature-controlled room on a 12-hour light-dark cycle (light period: 7am-7pm). The pregnant females or P4 litters with a dam assigned to the cuboid-reared group were housed in a standard, bedded 20 (width) x 45 (length) cm 'cuboidal' transparent, amber plastic home cage with ad libitum access to food on top of the cage and water. The difference between the length and width dimensions of the cuboidal home cage increased cage asymmetry providing additional means of orienting in the environment. These rats were reared in an experimenter-controlled isolated environment (using a plain, black curtain and vertical walls) to restrict any potential access to extra-cage visual cues. In contrast, the pregnant females assigned to the sphere-reared group were housed inside a 23-inch (57.5 cm) diameter white translucent plastic spherical home cage with bedding and ad libitum access to food placed on the cage floor and water delivered through an easily accessible waterspout entering the sphere through a single hole placed sideways, 10 cm-high above the bottom of the sphere. The spheres were composed of two identical hemispheres (domes) that were perfectly attached on top of each other at the level of sphere equator. Several 1-mm-diameter holes were made at -10 cm below the sphere top to allow for continuous air flow and exchange. All pregnant females were visually inspected 2–3 times a day without intervening with their ongoing behavior and the birthday of the litter was assigned as P0. Rat pups implanted before P21 were co-housed together with the dam and their littermates. All rats were weaned at P21. Following surgery, weaned rat pups were housed in a cuboidal home cage with dimensions 19 (width) x 37 (length) cm or a spherical home cage with a diameter of 15-inch (37.5 cm). Sphere-reared rats had no access (visual or physical) to geometric linearity for their entire lives until the first day of the experiment, as they were born in the sphere, and subsequently, great care was taken at each step before the experiment (i.e., anesthesia induction, sleep box, single-housing home cage) to prevent such access. While the sphere housing the rats for the experiment was kept in one spatial location throughout the experiment, if momentary movement of the housing sphere was required, indirect experience with linearity through passive motion was prevented by moving the sphere in curved/non-linear motions. Light ambience in the sphere- and cuboid-reared environments were kept similar between analogous stages of the experiment. Each day, on completion of the experiment, the sphere-reared rats were returned to a spherical home cage. This experimental protocol allowed us to quantify the effect of controlled exposure to geometric linearity per day. A total of 60 rats (21 sphere-reared rats: group behavior experiment: $N = 8$, at age P17 and P21, sex: both sexes; corticosterone experiment: $N = 10$, aged P25-26, sex: male; electrophysiology experiment: $N = 5$, aged P24-P27, sex: male; 2 of these rats participated in corticosterone experiment as well; and 39 cuboid-reared rats: group behavior experiment: $N = 7$, at age P17 and P21, sex: both sexes; corticosterone experiment: $N = 10$, aged P25-P26, sex: male; electrophysiology experiment: $N = 24$ aged P15-27 and adult with 2 of these rats participated in corticosterone experiment as well, aged P17 and P21, sex: male) were employed in the current study. All experimental procedures were approved by the Yale University IACUC committee and were performed in accordance with NIH guidelines for ethical treatment of animals.

### Rat group behavior in cuboid versus spherical home cages across development

To investigate the overall locomotion in a home cage setting, a camera inserted through a small hole at the top of the cuboidal (for cuboid-reared rats) and spherical (for sphere-reared rats) home cages was used to record the rat-group behavior and locomotion at the ages P17 (age at which normally-reared rats move freely) and P21 (age at which surgery was performed for the experiment) for the whole litter (cuboid: $N = 7$; sphere: $N = 8$ rat pups) with the mother for a 15 min session. Subsequently, multi-animal DeepLabCut software[20], a publicly available software package allowing simultaneous multi-animal location estimation developed in Python, was used to extract 200 video frames per age and group for hand annotation. A k-means clustering algorithm of the video frames was conducted to ensure the group of 200 frames provided a varied, diverse sampling covering different experimental scenarios. Seventeen points on the rats (spanning various points from head to body extremities) were hand-annotated on each frame for each rat (the dense labeling ensured accurate prediction of rat location in a group setting). Differences in rat fur patterns and distinct marks on the body (added using a marker) were used to consistently label each of the rat pups across the 200 frames. Next, a pretrained deep neural network in multi-animal DeepLabCut was trained with these frames separately for each age and group. Subsequently, the videos of the rat behavior were processed through this neural net to determine animal location. To quantify the movement in each of these cages, we computed the linearized distance covered in a movement bout when velocity was >5 cm/s and the number of such movement bouts/unit time for each of the rats.

### Fecal corticosterone quantification using enzyme linked immunosorbent assay (ELISA)

Rats born and reared in cuboidal or spherical home cages were first exposed to linear tracks on P25 and P26 (total: $N = 10$ rats/group; 2 of these rats/group followed the electrophysiology protocol detailed below). Dry fecal samples were collected while the rats explored the linear tracks, and fecal material was stored at $-20\,°C$. Fecal matter collected over 2 days during exposure to linear tracks from the same rat was combined to gain appropriate amount of fecal material/rat ($N = 10$ rats per group). Fecal corticosterone was measured using a well-validated commercial ELISA kit (Arbor Assays, catalog no. K014, Ann Arbor, MI, USA). The protocol provided by Arbor Assay was followed to extract corticosterone. 0.2 g of crushed, dry fecal material per rat was weighed. One mL of ethanol was added per 0.1 g of fecal material, then vortexed and shaken. Samples were subsequently centrifuged at 4 °C and the supernatant isolated, which was evaporated to dryness using a SpeedVac. The corticosterone ELISA kit from Arbor Assays was used for conducting the ELISA. Subsequently, absorbance was read at 450 nm on the BioTek microplate reader (BioTeK Instruments Inc., Vermont, USA).

### Surgery for implantation of neural probes

Electrode implantation surgery was conducted under 1–2% isoflurane general anesthesia provided intra-nasally. Appropriate depth of anesthesia was maintained by periodic, regular checks followed by minor adjustments to anesthesia levels. In the sphere-reared group, anesthesia was induced inside a small spherical chamber (12-inch diameter) into which they were transferred from the home cage under a dim red light, thus preventing any access to geometric linearity. During surgery, for both groups of rats, body temperature was maintained by a heating blanket. Neuronexus silicon probes (32 recording sites, 4 shanks x 8 sites/shank, Buz32 or 64-sites, 6 shanks x 10 sites/shank or 5 shanks x 12 sites/shank, Buz64) were attached to moveable microdrives and used for the current study. These were implanted bilaterally above each CA1 of dorsal hippocampus (64–128 total recording sites) at the following coordinates: AP of 3.0 mm posterior to bregma, ML of 1.8 mm lateral to the midline and DV of 1.5 mm ventral to brain surface. One cuboid-reared (P20 on Day1 of the electrophysiological recording) rat was implanted above the right hippocampus only (same coordinates) with a moveable 64-site Neuronexus silicon probe (Buz64). For ground and reference electrodes, jeweler's screws were implanted above the cerebellum posterior to lambda. To anchor the implant to the skull,

additional screws were used. In total, the head mount weighed approximately 4–5 g allowing free behavior of the rat pups. After recovery from the surgery, the rat pups with the head mount implant were fed soymilk formula for infants and returned to their respective home cages of corresponding geometry for overnight housing.

On the 2–3 days following the surgery, the neural probes were gradually moved down to the CA1 area of the dorsal hippocampus into the pyramidal layer (identified by 140–200 Hz ripple oscillations and by estimated electrode depth) while rats rested and slept in heated sleep enclosures of corresponding geometry. The small spherical sleep box had a hole at the top sufficient to allow the placement of the rat by the experimenter, the free movement of the recording tether and the video recording of the animal location by an overhead video camera. The sleep box used for the cuboid-reared group was a small, high-walled cuboid environment. In the sphere-reared rats, probes were lowered under a dim red light to prevent these animals from viewing environmental geometric linearities. On obtaining stable recordings at the optimal probe location, the experiment was commenced[10,11].

## Grouping of rats for electrophysiological recordings

Electrophysiological recordings were performed over 4 days (Days1–4) in the developing cuboid- and sphere-reared rats and distinct experimental protocols were followed on each day (detailed with rationale in the next sections and Figs. 1b, 5a, 7a) to investigate different properties of hippocampal neuronal ensembles. For the sphere-reared group, recordings were conducted on 5 sphere-reared male rats aged P24 on Day1 of the experiment. The age of P24 was selected because the emergence of adult-like hippocampal neuronal ensembles occurs at P23-24 in control, cuboid-reared rats[11]. Successful hippocampal neuronal ensemble recordings were additionally obtained from all these rats on Day2 of the experiment (P25) and 4 of these sphere-reared rats on Day3 (P26) and Day4 (P27) of the experiment (one rat's implant became unstable on Day 3 and recordings were stopped). For the control, cuboid-reared rats, first day of the electrophysiological recordings started at P15-24 or adult ages (Day1 of the experiment) and were conducted on 21 cuboid-reared male rat pups and 3 adult cuboid-reared male rats. To generate a developmental timeline of various properties of hippocampal neuronal ensembles in cuboid-reared rats, the data obtained in the developing cuboid-reared rats were grouped in two-day age groups of 3-5 animals per group starting at P15 (i.e., P15-16, P17-18, P19-20, P21-22, P23-24 on Day1). Additionally, electrophysiological recordings were obtained from 3 adult cuboid-reared rats were with the same Day1Run1 protocol. On Day2, we obtained successful electrophysiological recordings from a total of 19 cuboid-reared developing rats grouped as P16, P17-20, P21-23, P24-25 age-groups. Cuboid-reared rats aged P24-25 were grouped together on Day2 (N = 5). On Day 2, one out of the 5 P24-25 cuboid-reared rats had to be excluded from neuronal ensemble analyses (i.e., p/replay, ensemble clusters, cell assembly plasticity) due to low simultaneously recorded cell count. On Day3 (P25-26) of the experiment, ensemble recordings were obtained from 3 cuboid-reared rats, while on Day 4 (P26-27) recordings were performed from 3 rats (one animal's implant became unstable and no recordings were performed). During the experiments, to supplement the diet of the rat pups they were occasionally fed soymilk formula for infants.

## Experimental design for electrophysiological recordings on Day1

Activity of hippocampal neuronal ensembles was recorded while the untrained, naïve rat pups slept in a heated high-walled opaque cuboidal box of 20 × 20 cm for cuboid-reared rats and a heated opaque sphere (12 inch or 30 cm diameter) for sphere-reared rats for a duration of -1.5 h (Pre-Day1Run1 sleep). Next, the rat pups were placed on a novel linear track (Track 1), which was 1 meter long. The rat pups in both groups explored the linear track well without any training (de

novo run, Day1Run1) as they had no prior experience of linear tracks or other mazes. Therefore, while both cuboid-reared and sphere-reared rats had no experience of extended linear space, sphere-reared rats additionally had no experience of geometric linearities including linear borders, planarity, ends and corners in their spherical home cages or sleep boxes, which are naturally found in the cuboidal home cages and sleep boxes. The sleep boxes before and during the experiment for both groups were placed in a distinct spatial location from that of the linear tracks. Exploration of the linear track, Day1Run1, was followed by another sleep session of similar duration in the sleep enclosure of corresponding geometry (Post-Day1Run1 sleep) recorded in the same location as the Pre-Day1Run1 sleep. A fraction of the cuboid-reared rats from all age groups and all sphere-reared rats ran again on linear Track 1 (Day1Run2) after Post-Day1Run1 sleep. Subsequently, the rat pups were returned to their respective home cages of corresponding geometry. Three implanted adult cuboid-reared rats (with no prior training of linear tracks) went through a similar protocol to determine characteristics of hippocampal neuronal ensembles in adult cuboid-reared rats. These rats slept in a cuboidal sleep box for several hours, followed by a Run session on a 1.5-meter linear track for the first time in their lives. The Run was followed by another sleep session of comparable duration in the same sleep box.

## Experimental design for electrophysiological recordings on Day2

On experimental Day2, all rats went through a protocol aimed at studying the hippocampal neuronal representations for encoding and storage of multiple track experiences[10]. Since sphere-reared rats had no experience with 90-degree corners, the 3 tracks were partitioned by 90-degree corners, environmental landmarks absent inside spheres. Following a 1.5–2-h Pre-Day2Run1 sleep session, the cuboid- and sphere-reared rats first ran on the same linear track familiarized on Day1 (Track 1, Day2Run1), after which barriers blocking access to two novel tracks orthogonal to Track 1 (Tracks 2 and 3, 1-m-long each, U-shape maze) were lifted. After running on these 3 contiguous tracks (Day2Run2, U-shape maze), the animals slept (Post-Day2Run2 sleep) for -1.5–2 h in their respective sleep boxes of corresponding geometry and subsequently ran on all 3 tracks again (Day2Run3). Day2 experiment was ended by returning the rats to their respective home cages.

## Experimental design for electrophysiological recordings on Days3–4

On experimental Day3, to investigate how experience with geometric linearities might reshape the development of hippocampal neuronal ensembles, we designed the following experiment: after a -1.5–2 h sleep session (Pre-Day3Runs sleep) in sleep boxes of corresponding geometry, the rats sequentially ran on multiple, single 1-m-long tracks (i.e., the familiar Track 1 and novel Tracks 4 and 5, Day3Run1,2,3). The rats then slept in their respective sleep boxes (Post-Day3Runs sleep) and subsequently re-ran on Tracks 4 and 5 (Day3Run4,5).

On Day4 of the experiment, after a -1.5–2 h sleep session (Pre-Day4Runs sleep), rats were placed on one previously experienced (familiar) track (Track 4; third exposure, previous 2 exposures were on Day3), followed by exposure to novel Track 6. The rats then slept in their respective sleep boxes (Post-Day4Runs sleep) and subsequently re-ran on track 6.

For all recording days, to increase the chances of the rat pups sleeping between the run sessions (a cumulative total of 67 days of electrophysiological recordings from rats across groups and ages), the experiments were performed during the light phase of the day.

## Electrophysiological recordings

A Neuralynx digital recording system (DigiLynx with capability to record 128-channels simultaneously) was used for the electrophysiological recordings. During the experiment, a head-mountable

amplifier connected the rat to the recording system via a lightweight multiplexed cable. The cable was balanced by a sliding pulley system which enabled free rat movement. Tracking of rat position was enabled by two LEDs (red and green) 1.5 cm apart in the head stage amplifier captured by an overhead video camera. The DigiLynx system was used to simultaneously record wideband local field potentials (range: 1–6000 Hz), >50 microvolts putative spike waveforms (range: 600–6000 Hz), LEDs/animals position tracking coordinates and a video of the experiment. To confirm recording locations (i.e., the pyramidal layer of the dorsal hippocampal CA1) on completion of the electrophysiological recordings, transcardial perfusion of the rats with 0.9% saline followed by 4% paraformaldehyde was performed, and the brains were harvested, sectioned and followed by Cresyl violet Nissl-staining to aid visualization.

## Single unit isolation

The manual cluster-cutting software, Xclust3[10,26] was used to identify clusters in a multidimensional space composed of detected spike waveform amplitudes on nearby recording sites. Putative single-units were separated based on inter-spike intervals and cross-correlations across clusters. The single-unit autocorrelation, average firing rate and spike width were used to classify putative pyramidal units and inter-neurons. Further analysis was conducted only on well-isolated, stable, units (cumulative total of 3241 clustered single units across groups, days and ages from $N = 29$ rats).

## Unit cluster quality determination

To determine the quality of isolated spike clusters quality, the separation of putative spikes belonging to a cluster with regard to all other activity on its respective tetrode was computed using the isolation distance metric[62]. This metric was calculated separately for each session (Pre-Run sleep sessions, Run sessions and Post-Run sleep sessions) for each day to compare cluster isolation quality across sessions within a day. Omnibus Kruskal–Wallis ANOVAs were performed to determine if cluster quality varied with session type (Pre-Run sleeps, Run sessions, Post-Run sleeps) or across days of experiment (Days 1, 2, 3 or 4), while two-sided rank-sum tests were performed to determine if cluster quality differed by group identity (sphere-reared vs. cuboid-reared),

## Characterization of place cells

For each putative recorded pyramidal unit, the firing rate in each spatial location was determined by counting spike numbers (rat velocity > 5 cm/s) in non-overlapping 2 cm bins smoothed by a 2 cm Gaussian kernel and dividing these values by the occupancy (time spent in corresponding spatial bin at velocity > 5 cm/s, smoothed by a 2 cm Gaussian kernel). A maximal/peak firing rate threshold of 1 Hz was applied to characterize putative pyramidal neurons as place cells[10,11].

## Single cell firing characteristics

Various metrics of hippocampal place cells were computed to quantify tuning/firing characteristics of place cells on linear tracks in cuboid- and sphere-reared rats. (1) Peak firing rate: the maximal firing rate of a place cell on a track. (2) Spatial information (SpInfo): the extent to which an individual place cell's firing conveys information about space. It was computed as follows:

$$SpInfo = \sum_{j=1}^{L} Pr\left(loc_j\right) \frac{fr_j}{fr_{mean}} \log_2 \left(\frac{fr_j}{fr_{mean}}\right) \qquad (1)$$

where $fr$ is the firing rate of the cell in location $j$, $fr_{mean}$ is the mean firing rate of the cell, and $Pr(loc_j)$ is the probability of occupancy of location $j$. (3) Stability of each place cell within a Run session: the Spearman's correlation of the place field maps derived from the first and the last one-fifth of laps during the Run session. (4) Length of the primary place field: the total continuous/contiguous distance between

locations in space where the firing rate of the putative primary place field (with peak firing rate above 1 Hz) first/last exceeded 20% of the peak firing rate threshold during locomotion (velocity > 5 cm).

## Bayesian decoding of spatial location on the track during Run

To study the behavioral-timescale representation of space by neuronal ensembles, a memoryless Bayesian decoding algorithm[22,26] was applied to the activity of all place-responsive pyramidal cells (with at least 10 spikes during Run, binned at 1 cm and smoothed by a Gaussian kernel of 5 cm) on the track in non-overlapping time bins of 0.5 s when rat velocities exceeded 10 cm/s[10]. Each linear track run direction was separately and independently analyzed as in our prior work[10,11,16]. In brief, Bayes' theorem states,

$$Pr(loc|spk) = \frac{Pr(loc)Pr(spk|loc)}{Pr(spk)} = \frac{Pr(loc)Pr(spk|loc)}{\sum_{j=1}^{L} Pr\left(loc_j\right) Pr\left(spk|loc_j\right)} \qquad (2)$$

where $Pr(loc|spk)$ is the posterior conditional probability of location given spikes, $Pr(loc)$ is the prior probability of location, $Pr(spk|loc)$ is the probability of spikes given a location, $Pr(spk)$ is the probability of spikes, $loc_j$ is the $j^{th}$ location on the track out of a total of $L$ locations.

Under the assumptions that spikes have a Poisson distribution and that cells have statistical independence[22]:

$$Pr(spk|loc) = \prod_{i=1}^{n} Pr(sp_i|loc) = \prod_{i=1}^{n} \frac{(\tau f_i(loc))^{sp_i}}{sp_i!} e^{-\tau f_i(loc)} \qquad (3)$$

Insertion of the above equation into Bayes' theorem, gives:

$$Pr(loc|spk) = Constant(\tau, spk) Pr(loc) \left(\prod_{i=1}^{n} f_i(loc)^{sp_i}\right) e^{-\tau \sum_{i=1}^{n} f_i(loc)} \qquad (4)$$

where $Constant(\tau, spk)$ is a normalization factor such that $\sum_{i=1}^{Pn} Pr\left(loc_i|spk\right) = 1$, where $f_i(loc)$ is the value of the smoothed averaged place-responsive cell firing of the $i^{th}$ place-responsive cell at a location, $sp_i$ is the number of spikes fired by the $i^{th}$ place-responsive cell in the time bin being decoded, $\tau$ is the duration of the time bin (0.5 s during locomotion on the track and 0.02 s for sleep frames) and $n$ is the total number of place-responsive cells. $Pr(loc)$, the prior for location, was assumed to be uniform across the linear track.

For each time bin with spikes during Run, the difference between the location with maximum Bayesian decoded probability from the true animal location was computed and considered as the decoded location error. Five hundred shuffled datasets were generated to determine if neuronal ensembles decoded the rat location above chance. These shuffles were generated by randomly permuting in time (500 time-bin shuffles) the decoded spatial posteriors. Subsequently, the decoded location error for shuffles were computed and compared with that of the original data.

## Peak theta frequency

Theta frequency was calculated while rats were moving at velocity > 10 cm/s using recording channels in the CA1 area with place cells, calculating the Hilbert Transform followed by a Fast-Fourier Transform of the concatenated epochs of movement, and finding the peak frequency within the theta range (5–9 Hz).

## Theta sequences nested in theta oscillations

Theta sequential structure was determined using only theta cycles with rat velocity > 10 cm/s, activity of a minimum of two neurons and while the rat was pup in the middle 2/5th (40 cm) of the linear track (>30 cm from the track ends of the 1-meter linear track). Bayesian decoding on the spike trains of the neuronal population was conducted in a 400 ms window centered on each theta

oscillation trough to determine the decoded virtual rat location based on the neuronal ensemble activity in 20 ms time bins (non-overlapping time bin window). For analysis of theta-sequential structure, a 1-cm spatial bin smoothed by a Gaussian kernel of 2 cm was used[11]. For every theta cycle, the decoded probabilities centered on the rat location on the linear track (± 30 cm) were isolated and averaged across all theta cycles to reveal the difference between the true and the predicted, decoded locations. The quadrant ratio was computed using a 100 ms time window centered on the trough of the averaged theta cycle. To test if any finer timescale (than the time bin) differences might exist across the groups, a similar analysis was performed using a 20 ms time bin shifted by 5 ms. Second, to test the impact of track ends on theta sequences we truncated the ends of the track (0–30 cm from one end at a time) and computed the average theta sequential structure in this reduced dataset as in our original dataset using non-overlapping time bin windows. Next, to study theta sequential structure at the individual theta cycle level, theta cycle frames with a minimum of 5 distinct neurons (with 20 ms non-overlapping time bin windows) were analyzed and compared to their respective time bin shuffles.

### Theta sequence quantification via the quadrant ratio

The quadrant ratio was computed by splitting the space-time decoded matrix centered on the rat into 4 quadrants. Next, the decoded probabilities in quadrants ahead (quadrant 1, locations visited in the immediate future) and behind (quadrant 3, locations visited in the immediate past) the current location of the rat in space and time were summed and the probabilities from the other two quadrants (quadrants 2 and 4) subtracted from their sum. Finally, this value was normalized by the total sum of probabilities in quadrants 1–4. The quadrant ratio evaluates the degree of forward-going theta sequential structure in the data centered on the current location of the rat. Theta sequence significance was determined by comparing with 500 shuffled datasets. These shuffled datasets were generated by randomly permuting the time bins and computing the quadrant ratio for the shuffled datasets. A group of rats had significant theta sequences if the group quadrant ratios were significantly above the 95th percentile of their respective shuffles. For individual theta cycle frames, the quadrant ratio of the frame was compared with that of 500 time-bin shuffles of that frame. A quadrant ratio exceeding the 95th percentile of the shuffled distribution was considered an individual frame with significant theta sequence.

### Frame detection during non-rapid-eye movement (NREM) sleep in the sleep box (cuboid or sphere)

Frames were detected during NREM sleep periods in the sleep boxes using criteria matching those used in our prior work, described before[10,11,16]. Briefly, using the summed population activity of all pyramidal neurons in 1 ms bins was convolved with a Gaussian kernel of 15 ms. Time epochs in NREM sleep when the population activity of at least 5 distinct neurons surpassed 2 standard deviations above the mean population activity for 100–800 ms were found and used as frames. NREM sleep was isolated using the following criteria[10,11,16]: rat immobility (rat velocity < 2 cm/s, set to this value since rat pups occasionally move/twitch during sleep), a low theta/delta ratio (theta/delta ratio below the mean, computed on the Hilbert transform for the respective frequencies, 5–9 Hz for theta and 1–4 Hz for delta, and smoothed with a 10 s Gaussian) to remove sleep REM epochs, combined with manual inspection and curation, if required. The frames in NREM sleep were binned at 20 ms time bins to allow accurate depiction of location within individual time bins[16]. The standard minimum frame duration of 100 ms was used to include at least 5 time bins (20 ms/bin) within a frame (necessary for appropriate statistical comparisons with shuffles). Subsequently, Bayesian decoding of population spike trains

within the 20 ms time bins was performed to determine the virtual location of the rat during NREM sleep. Each run direction was considered independent and analyzed separately for this analysis.

### Frame detection during awake rest on the track

Frames of heighted population activity were detected during awake rest epochs while the rats were on the linear track. The criteria used for frame detection during NREM sleep were applied on the run session as well to isolate epochs of awake rest with one difference: the threshold for rat velocity < 1 cm/s to avoid contamination from activity associated with active movement.

### Weighted correlations for quantification of frame trajectory sequential structure

Frame sequential content was quantified using a product-moment linear correlation between time and location[16], weighted by the associated posterior probabilities[16], computed as follows:

First, the weighted mean for location ($m_{loc}$) and time ($m_t$) was computed:

$$m_{loc}(loc; Pr) = \frac{\sum_{i=1}^{T}\sum_{j=1}^{L} Pr_{ij} loc_j}{\sum_{i=1}^{T}\sum_{j=1}^{L} Pr_{ij}} \tag{5}$$

Weighted mean for time was computed in a similar manner. This was followed by computation of the weighted covariance (*covar*):

$$covar(loc, t; Pr) = \frac{\sum_{i=1}^{T}\sum_{j=1}^{L} Pr_{ij}(loc_j - m_{loc}(loc; Pr))(t_i - m_t(t; Pr))}{\sum_{i=1}^{T}\sum_{j=1}^{L} Pr_{ij}} \tag{6}$$

And then, the weighted correlation *(r)*:

$$r(loc, t; Pr) = \frac{covar(loc, t; Pr)}{\sqrt{covar(loc, loc; Pr)(covar(t, t; Pr))}} \tag{7}$$

where $loc_j$ is the $j^{th}$ spatial bin, $t_i$ is the $i^{th}$ temporal (20 ms) bin in the frame, $Pr_{ij}$ is the Bayesian posterior probability for the $j^{th}$ spatial bin at the $i^{th}$ temporal bin, $T$ is the number of temporal bins and $L$ is the number of spatial bins. Note that $\sum_{j=1}^{L} Pr_{ij} = 1$ and $\sum_{i=1}^{T}\sum_{j=1}^{L} Pr_{ij} = T$.

### Shuffled datasets for decoded trajectory sequences

To determine if detected frames during theta, awake-rest or sleep significantly depicted the sequential run trajectory, 500 Monte Carlo shuffles were generated for each frame by randomly permuting the 20 ms time bins (containing the spatial decoding probabilities) within a frame (time bin shuffle). This within frame shuffle completely conserved the structure of decoded probabilities within each 20 ms time bin of that frame but randomized their sequential relationship with each other. This shuffle is immune to the false-positive or false-negative results for decoded trajectory sequence analysis induced on comparisons with other shuffle procedures, which were shown to be due to lack of conservation of other essential characteristics of the data in those shuffles. These include shuffles that circularly-shift decoded probabilities in a 20 ms time bin[16], or the cell ID shuffle[27], that can potentially lead to spurious conclusions.

### Proportion of frames with trajectory sequences during theta

The proportions of significant trajectory sequences were calculated by comparing the quadrant ratio or weighted correlation of each individual frame with its respective shuffles (time bin shuffle). A frame was considered significant if the weighted correlation was above the 95th percentile of its respective shuffles. The proportion of significant frames was calculated by dividing the number of significant frames by the number of detected frames in that run session. We conducted

3 separate statistical tests to determine if the proportions of significant theta sequence frames were above chance: (1) a statistical comparison with significant proportions in a time-binned shuffled dataset, (2) a binomial test against 0.05 chance, (3) a statistical comparison with the absolute 0.05 chance value.

### Proportion of frames with trajectory sequences during awake-rest or sleep

To determine the proportions of frames with significant trajectory sequences, the weighted correlation of each frame was compared with its respective time bin shuffles. Frames depicted significant run trajectory sequential content if the weighted correlation was above the 97.5th percentile (for forward p/replay sequences) or lower than the 2.5th percentile (for reverse p/replay sequences) of the shuffles. The number of significant frames were then divided by the total number of detected frames in that sleep/awake rest session to compute the proportion. In order to determine if proportions of significant trajectory sequences in the data were above chance, we conducted the following 3 separate statistical comparisons: (1) we compared the proportions of observed trajectory sequences in the data with the proportions detected in a time bin shuffled dataset undergoing the same procedure, (2) we computed a binomial test for each group (chance level set to $p = 0.05$), (3) we compared the proportion of significant frames per rat run direction for the group with the absolute 0.05 chance value.

### Proportion of frames with significant track-specific preplay for novel tracks

For Run sessions with exposure to multiple novel tracks (on Day2), we computed track-specific preplay during the sleep preceding the Run sessions on novel tracks. Significant frames for each track were identified including p/replay for tracks with prior exposure (e.g., track 1 in Pre-Day2Run sleep), and the total number of frames significant only for one, not the others, of the multiple experiences were counted. This value was divided by the total number of sleep frames to obtain the proportion of frames with track-specific preplay for novel tracks. Multiple statistical comparisons (i.e., t-test vs. a time-bin shuffled data, Binomial test vs. $p = 0.05$ chance, t-test vs. $p = 0.05$ chance) were performed as in the previous section to determine significance of such sequences. Note that the time-bin shuffled frames also underwent the same procedure and had to exceed the same two criteria (1) exhibit significance for that particular track, and (2) in addition, exhibit non-significance for other tracks.

### Proportion of frames with significant track-specific preplay for future tracks

For Run sessions with exposure to multiple novel tracks (on Days3–4), we computed track-specific preplay during the sleep preceding the Run sessions for all tracks in that run session. Significant frames for each track were identified, and the total number of frames significant only for one, not the others, of the multiple experiences were counted. This value was divided by the total number of sleep frames to obtain the proportion of frames with track-specific preplay for future tracks, followed by the same statistical comparisons detailed in the previous section.

### Sequence score for quantification of frame sequential structure

To normalize the influence of superfluous differences across frames (e.g., firing rates) and assess trajectory sequential structure, the sequence score ($r_Z$) was defined as the Z-score of the absolute weighted correlation of a frame relative to the absolute weighted correlations of its respective shuffled frames[16]. It was computed as follows:

$$r_Z = \frac{|r| - mean(|r(shuffles)|)}{std(|r(shuffles)|)} \qquad (8)$$

where $r$ is the weighted correlation of the frame and $r(shuffles)$ the weighted correlations of its respective shuffles.

### Median jump distance

To determine how continuous the depiction of linear space was by trajectory neuronal sequences in frames, we computed the median jump distance for each frame as follows: during the frames, the peak decoded virtual location was computed for each time bin. The median jump distance of a frame was the median of the distribution of ratios between the absolute value of differences between peak decoded locations in consecutive time bins and the length of the track, as in ref. 11.

### Simultaneous comparison of sequence score and median jump distance

Comparing multiple features of trajectory sequences provide a more detailed description of their sequential content. Therefore, we simultaneously examined the two features enabling most differentiation between sleep frames from each group and their respective shuffles (e.g., Pre-Day1Run1 sleep sphere-reared group with 500 time-bin shuffles). The two features examined for all sleep frames were the sequence score and median jump distance as in ref. 11. To simultaneously examine frame sequence score and median jump distance for a sleep session, thresholds for sequence score in an increasing fashion (higher shuffle-normalized trajectory sequential content) and thresholds for jump distance in a decreasing fashion (less jumpy from one 20 ms time bin to the next) were independently set, and the proportion of frames (out of all detected frames) passing both thresholds were computed. The values set for the feature thresholds were as follows, for sequence score: no threshold (i.e., all frames), 0 to 2.8 in steps of 0.4 for sequence score, and for median jump distance: 0.125 to 1 in steps of 0.125 between thresholds. For comparison with shuffles, sequence scores and median jump distances for a shuffled dataset were computed similarly to the data, and a one-sided Z-test for 2 proportions performed between the data and shuffle for each set of thresholds (72 sets of total thresholds). The test Z-score for each threshold set was converted to a $p$-value. Statistical significance was set at $p < 0.05$. A pseudo-colored matrix was used to visualize these p-values.

Additionally, although we presented results for all pairs of thresholds, we also identified two pairs of thresholds as more meaningful for sequence comparison based on empirical assessment of the results. These pairs of thresholds were as follows, thresholds 1: sequence score $> 1.60$ and median jump $< 0.375$; thresholds 2: sequence score $> 1.60$ and median jump $< 0.25$. These two pairs of thresholds were set to ensure comparisons of frames across conditions had sufficiently low jump (were continuous frames), sequentially-depicted a significant proportion of the track experience (were not stationary) and had sufficient frames in each condition.

### Maximum jump distance

To ensure that only trajectory-depicting neuronal sequences lacking a large jump in trajectory depiction were present in the data, the peak decoded virtual location was computed for each time bin within the frames. The maximum jump distance of a frame was the maximum of the distribution of ratios between the absolute value of differences between peak decoded spatial locations in consecutive 20 ms time bins and the length of the track. While this is a biased estimator of trajectory representation[11] as longer-duration frames are more likely to probabilistically have a large maximum jump distance, this analysis was conducted to rule out the possibility of frames with large maximum jump distance exhibiting p/replay[16]. Longer-duration frames, generally having a higher proportion of activated place cells, carry the

most spatial trajectory-related sequential content and are crucial for mnemonic purposes[16,33].

## Simultaneous comparison of weighted correlation and maximum jump distance

A simultaneous comparison of two additional features between sleep frames from each group and their respective shuffles (e.g., Pre-Day1Run1 sleep in the sphere-reared group with 500 time-bin shuffles) was performed for each of the two groups (cuboid- or sphere-reared rats), sessions in each of the sleep sessions (Pre-Day1Run1 sleep or Post-Day1Run1 sleep). The two features examined for all sleep frames were the weighted correlation and maximum jump distance. To simultaneously examine weighted correlation and maximum jump distance of frames for a sleep, thresholds for weighted correlation in an increasing fashion (higher values indicate higher trajectory sequential content) and thresholds for maximum jump distance in a decreasing fashion (lower values indicate a lack of high maximum deviation from one time bin to the next) were independently set, and the proportion of frames (out of all frames) passing both thresholds were computed. The threshold values for these features were as follows, for absolute weighted correlation: 0 to 0.9 in steps of 0.1, and for maximum jump distance: 0.1 to 1 in steps of 0.1 between thresholds. The percentile of the proportion in data relative to the shuffled datasets (e.g., Pre-Day1Run1 > 500 time-bin shuffled datasets) for each set of thresholds, was expressed as a $p$-value by subtracting the value from 1. Significance was set at $p < 0.05$. A pseudo-colored matrix was used to visualize the p-values.

Additionally, although we presented results for all pairs of thresholds, we also identified a set of thresholds as more meaningful for weighted correlation comparison. These thresholds are weighted correlation > 0.6 and maximum jump distance < 0.4 (depicted using a red rectangle in the p-value matrix). While there remains a degree of arbitrariness to setting these thresholds, the considered frames on using these thresholds would have high sequential content and low maximum jumps.

## Generation of a single surrogate Poisson dataset for comparison with real data

To study the impact of average sleep firing rate differences across neurons on the observation of trajectory sequences in cuboid- and sphere-reared rats, the averaged (mean) firing rate of each neuron across all frames in Pre-Day1Run1 sleep or Post-Day1Run1 sleep was first computed. Next, a Poisson dataset was generated with each neuron having matched average (mean) firing rate to the data, but random spike firing times by drawing spikes via a homogenous Poisson process. The proportion of frames passing increasingly stringent thresholds for sequence scores (normalized by $n = 500$ time-bin shuffled datasets) and median jump distances were simultaneously compared between the real data and surrogate, rate-matched Poisson dataset in each group.

## Generation of 500 surrogate Poisson datasets for direct comparison with real data

To study the impact of average firing rate differences across neurons on the observation of trajectory sequences, 500 Poisson datasets were generated with each neuron having matched averaged (mean) firing rate to the data, but random spike firing times generated by drawing spikes via a homogenous Poisson process[16]. The proportion of frames passing increasingly stringent thresholds for absolute weighted correlation and maximum jump distances were simultaneously compared between the real data directly with 500 surrogate, rate-matched Poisson datasets in each group. While the analysis described in the previous section (data vs. single shuffle-normalized Poisson dataset) allows a direct comparison of excess sequential structure compared to their respective shuffles

(assessed via the sequence score), this analysis directly compares the data with 500 rate-matched Poisson datasets. Our results were similar using both procedures.

## Trajectory sequence characterization after controlling for differences in spatial information

To account for the group differences in average spatial information, we performed two analyses. First, to test the impact of cells with lowest spatial information in sphere-reared rats on p/replay, we dropped cells with low spatial information in the sphere-reared rats until both cuboid- and sphere-reared rats had similar spatial information as assessed by a Wilcoxon's rank-sum test. Subsequently, we performed the same sleep analysis to study preplay, replay and plasticity in this spatial information matched dataset. Second, to test the impact of spatial information differences across groups, we dropped cells with high spatial information in cuboid-reared rats and low spatial information in the sphere-reared rats until both cuboid- and sphere-reared rats had similar spatial information as assessed by a Wilcoxon's rank-sum test. Subsequently, we performed the same sleep analysis to study preplay, replay and plasticity in this spatial information matched dataset.

## Hippocampal high-frequency ripple oscillations

The local-field potential was filtered in the 140–250 Hz range to find hippocampal ripples. Individual ripples were characterized as time epochs with maximal ripple power exceeding 3 standard deviations of the mean ripple power in this range. Trajectory sequential content in p/replay was then studied in sleep frames coinciding with hippocampal ripples.

## Detection of significant location-depicting cell assemblies

For cell assembly detection[16,30], the activity of place-responsive cells during movement in the run session was divided into 20 ms time bins. Each cell's activity for a particular run direction was Z-scored to limit bias towards high firing neurons:

$$z_{i,t} = \frac{spk_{i,t} - \mu_{spk_i}}{\sigma_{spk_i}} \tag{9}$$

where $spk_{i,t}$ was the spike count of neuron $i$ in bin $t$, and $\mu_{spk_i}$ and $\sigma_{spk_i}$ were the mean and standard deviation of neuron $i$'s spike counts across bins, respectively.

If we let $Z$ be the matrix of size $n$ (number of neurons) by $M$ (number of time bins), with element $(i,t)$ equal to $z_{i,t}$ then for this matrix, constituting the z-scored binned activity of the population of cells, cell assemblies were computed following the procedure outlined below.

A principal component analysis (PCA) was done on matrix $Z$:

$$\sum_{j=1}^{n} \lambda_j p_j p_j^T = \frac{1}{n} ZZ^T \tag{10}$$

where $p_j$ is the $j^{th}$ principal component with corresponding eigenvalue $\lambda_j$, and $\frac{1}{n}ZZ^T$ is the correlation matrix of $Z$, while $Z^T$ is the transposed version of $Z$. Next the Marcenko–Pastur law[30] was used to estimate the number of significant cell assemblies in the data as follows: this law postulates that for a $n$ by $M$ matrix whose elements are random variables which are independent, identically distributed, have a mean of zero and unit variance, all the eigenvalues are bounded asymptotically to the interval $[(1 - \sqrt{n/M})^2, (1 + \sqrt{n/M})^2]$, i.e., when $n$, $M$ approach infinity, $M/n$ converges to a finite positive value. Thus, this suggests that if neuronal firing rates are independent from each other, then eigenvalues are expected to not exceed $\lambda_{max} = (1 + \sqrt{n/M})^2$. Hence, only principal components with eigenvalues exceeding $\lambda_{max}$ were defined as significant cell assemblies.

## Activation of significant run cell assemblies in sleep

Strength of activation of cell assemblies in sleep neuronal activity were calculated by first binning the activity of each cell at 20 ms and z-scoring the binned vector. The extracted weights vector for each significant cell assembly/principal component was used to calculate its sleep activation strength as follows: the outer product of each weight vector was computed to derive a projection matrix, $P_k$. The diagonal of the projection matrix was zeroed to prevent erroneous high activation strengths caused by single neuronal activity. A quadratic product of projection matrix and instantaneous normalized firing rate of the neurons to compute the activation strength of each assembly.

$$A_c(t) = z(t)^T P_c z(t) \tag{11}$$

where $A_c(t)$ is the activation strength of cell assembly $c$ at time $t$ and $z(t)$ is a vector containing the normalized firing rate of all the neurons at time $t$.

To compute cell assembly activation strengths within Pre-Run sleeps and Post-Run sleeps, cell assembly activity in the NREM sleep epochs (binned in 20 ms windows) were used. A threshold of 5 with a minimum of two active neurons (including a minimum of one place cell) was used to detect significant cell assembly activation, and all instances of such activation were noted and averaged for each assembly. For each of the 4 days, the plasticity of these cell assemblies was studied. Cell assembly reactivation (or plasticity) was defined as average activation in Post-Run sleeps minus activation in Pre-Run sleeps for each assembly. To determine if an individual cell assembly was significantly affected by experience, a Wilcoxon's rank-sum test was conducted between all its instances of its activation in the Pre- and Post-Run sleep.

## Visualization of the neural manifold during Run on an extended linear track

To visualize the geometric structure of place cell population activity (i.e., the neural manifold for an extended linear experience), the place maps for all simultaneously recorded place cells were used. A min-max normalization for each spatial bin was performed to attenuate the effect of population differences in rate across the track while conserving correlations across spatial bins. The firing rate of the maximally active cell in a spatial bin was set to 1 and the remaining neurons scaled accordingly (from 0 to 1). Next, a principal component analysis was run on the population place cell activity maps. The first most informative 3-dimensions of the PCA accounted for 60-70% of the variance in the data, while 8 dimensions were required on average to account for 95% of the variance in each of the two groups. The neural trajectory for the population was plotted in the first 3 most-informative dimensions of the principal component space only to visualize the similarity of various segments of the trajectory in cuboid- and sphere-reared rats on a 1 m-long linear track. Note, the differences between groups in lower dimensions (4−8) were less marked; however, it is important to consider that those dimensions are also less informative as well.

## Mapping of linear space by neuronal ensembles during Run

All place field maps for a track were stacked, and the population vector for each location (using a 2 cm spatial bin) on the track was correlated with all locations on the track and the resultant matrix depicted as a heatmap. For quantification of similarity of the track ends, the locations within 15% of track length at each of the track ends were used. The average place field map population vector correlation for all these selected bins between the two track ends was computed to determine the similarity between these segments of the experience. Each direction of motion was separately analyzed and considered independent. Thus, 2 average correlation values were obtained per rat per track for comparisons between each of these segments and were compared between cuboid- and sphere-reared groups.

To study altered representation of linear space, population vector correlations of place cells were computed between equidistant (symmetrical) segments from the middle of the track. Each segment consisted of two spatial bins (bin size: 2 cm). The last segments composed of two bins at the ends were not considered to remove the observed effect of differences at track ends. We computed a Pearson's correlation between the distance between locations in each segment and their corresponding population vector correlations. This correlation was computed on locations 24 to 88 cm apart to remove bias induced by similar representation of very close by locations (<24 cm apart). Population vectors for more distant locations in linear space are naturally more dissimilar, leading to a negative correlation or a non-significant correlation (in the scenario where all studied locations are dissimilar to each other, as is the case on increased experience of a linear track in cuboid-reared rats). If spatial representation is changed in a sphere-reared rat, locations further apart would have more similar population vectors leading to a significant positive correlation. We also directly compared the population vector correlations between the two groups for each track segment (using two-sided rank-sum tests). In addition, we calculated the slope of the line of best fit (using least-squares linear regression) to estimate the degree of change in correlation values for a 1-m track. For all these analyses, either the correlations and linear regressions were computed on the average population vector correlations for each pair of symmetric locations across all directions and rats (group effect) or were separately computed for each direction of motion for each rat (individual direction/animal results).

To test if changed representation of linear space in sphere-reared rats was due to worse tuning of place cells, we dropped cells with low spatial information or high primary place field length in sphere-reared rats. Following each of these procedures, the analysis for mapping of linear space (described in the previous paragraph) was repeated.

## Classification of well-tuned, low-tuned and symmetric-mapping individual place rate maps during Run on linear tracks

We characterized place maps exhibiting lower place tuning, including rate maps with symmetric firing across the track about the track middle. Since firing rate of these symmetric rate maps was split across the two track halves, these cells exhibited lower spatial information for linear tracks (which prevented the usage of spatial information alone to isolate this rate map type). Thus, we used a two-step procedure to identify the proportion of different rate map types on rearing rats in cuboidal- or spherical- home cages. First, we identified rate maps which exhibited low tuning to linear tracks by quantifying the degree of variation of rate across the linear track. These rate maps were identified by first computing the ratio of squared-mean to variance (SMV) of each rate map across the track as follows:

$$\text{SMV} = \frac{[E(fr_i)]^2}{Var(fr_i)} \tag{12}$$

To identify low-tuned rate maps the SMV was computed for adult cuboid-reared rats. Rate maps with SMV above 99th percentile of adult place maps during a de novo run on a linear track (SMV > 3.62) were identified as having low tuning for linear space (or non-adult-like tuning; the observed results, i.e., sphere-reared rats had a greater proportion of low tuned rate maps than the age-matched cuboid-reared rats, were robust to a range of parameter values: from 2.5 to 4.0). The total proportion of such rate maps was computed. In the second step, out of the remaining population, a symmetry index (SI) was computed, to isolate neurons with symmetric rate maps from those which have good tuning for linear space as follows:

$$\text{SI} = 1 - \frac{PF - PF_{\text{sym}}}{PF + PF_{\text{sym}}} \tag{13}$$

where PF is the average firing rate in the 5 spatial bins centered on the peak firing location on the track for a cell, and $PF_{sym}$ is the average firing rate in the 5 spatial bins centered on the symmetric location across the track. Thus, a higher SI would indicate higher symmetric firing on the linear track. Rate maps with SI > 0.67 (the difference in average rates of identified symmetric locations across the two track halves was less than only 1/3rd the sum of rates across these locations) were identified as exhibiting symmetric rate maps (the observed results, i.e., sphere-reared rats had a greater proportion of symmetric rate maps than the cuboid-reared rats, were robust to a range of parameter values from 0.6 to 0.75) while the remaining rate maps were classified as rate maps well-tuned for linear space. To avoid characterizing rate maps spanning the track middle as symmetric (due to their peak firing location being close to the track middle), this analysis was limited to the rate maps with peak firing rate within 30 cm from one of the two track ends (60 cm in total of the 1 m track).

## Track end specificity in decoded sleep frames/trajectory sequences

We observed cases of trajectory replays in Pre- and Post-Day1Run1 sleep with track end depictions (or occasionally other symmetric locations furthest apart within a trajectory sequence) swapped, but part of remaining sequential experience faithfully p/replayed in sphere-reared rats. To study this phenomenon, we performed four different analyses (three of which are detailed in this section and one in the next section). To determine if the presence of such trajectory sequences with swapped starts and ends (or occasionally other symmetric locations) account for the differences in p/replay between cuboid- and sphere-reared rats we performed multiple analyses. First, we removed time bins (0, 1, 2, or 3 bins) from either the start or the end of the frames and determined if the frame significantly depicted a linear trajectory relative to its respective time bin shuffles. The difference in proportion of significant frames in pooled data across rats per group (providing a conservative estimate of the impact of this procedure on the difference) was computed for preplay and replay in the original data (removal of 0 bins). Subsequently, on removal of 1–3 time bins the proportion of new frames which became significant were added to the original significant frames and the difference between cuboid- and sphere-reared rats computed again. Only frames with 5 or more time bins after removal of the frame start or end were considered for sufficient statistical power. A Z-test for 2 proportions was used to compare the proportions of pooled cumulative significant frames across all rats in cuboid- and sphere-reared rats. As an additional control, 1–3 time bins were removed from the middle of the frames using the same procedure, and the resultant difference in cumulative proportion of significant frames computed between cuboid- and sphere-reared rats.

Second, to test the impact of track ends on track end swapping in sleep frames within trajectory sequences, we removed place maps' activity from the ends of the track (0–30 cm from one end at a time) and computed significant trajectory p/replay sequences in this reduced dataset as in our original dataset. This method elucidates the impact specifically of the two track ends on trajectory sequences, while the former approach (removing time bins from frame starts or ends) studies the impact of 'swapping' amidst track ends or other symmetric locations as well. Subsequently, we compared the differences in p/replay across groups in the original datasets versus the reduced dataset to study the impact of this procedure. As a control, we removed place maps' activity from the middle of the track (0–30 cm) and computed significant trajectory p/replay sequences in this reduced dataset as in our original dataset.

Third, instead of the linear-linear weighted correlation between space and time of decoded frames, we computed the circular-linear weighted correlation for each frame assuming space is circular (i.e., wraps around on itself, rather than linear) for the decoded frames.

These correlations were computed for their respective 500 time-bin shuffles as well. We subsequently followed the same procedure for analysis of the dataset and compared the differences across groups in p/replay using linear versus circular statistics.

## *T-1* time-space contingency analysis for track end depiction in sleep or awake-rest frames

To further study the track end depiction within decoded sleep frames, we focused on the short-timescale temporal relationships of decoded locations in frames. For each 20 ms time bin in a sleep or awake-rest frame its peak decoded location was determined. All time bins with the same peak decoded location across all sleep frames were grouped. The decoding probability across space for each of their preceding time bins (*T-1* time bin) of these bins was averaged. If sleep frames exhibit consistent, average sequential structure, this time-space contingency analysis would reveal a higher probability near the diagonal, which was our observation in cuboid-reared rats. To determine if the representation of track ends were swapped with each other in sphere-reared rats, a specificity index of decoded track ends was computed on the contingency matrix. The specificity index of decoded track ends was defined as the ratio of the average decoded probabilities on one end (15 cm) of the track (the actual predicted location) and the other end of the track (15 cm). A higher ratio depicts accurate short-timescale sequential depiction of track ends within frames, while a lower ratio is indicative of swapping of representations across the two track ends in consecutive 20 ms time bins within frames. As a control for non-specific differences across other nearby track locations with the track ends, we compared the specificity of the middle (15 cm centered on the track middle) with the two ends (15 cm) across groups.

## Cell pair correlations across sleep frames

For frames during sleep, the number of spikes (greater than or equal to 0) within a frame were counted for each of all frames. For all possible unique pairs of pyramidal cells, the Pearson's correlation coefficient was computed for these vectors to derive the cell pair correlations.

## Shuffle-normalized cell pair correlations across sleep frames

To remove any influence of firing rate of neurons across frames on cell pair correlations, spike counts/frame for each of the two neurons in the cell pair were randomly assigned new frame IDs to generate a frame ID shuffle. A Pearson's correlation coefficient between the two shuffled correlation vectors was computed. This procedure was repeated 500 times. The real cell pair correlation was then normalized by the shuffles by computing its percentile (expressed from 0 to 1) relative to its shuffles.

## Equalizing spatial information or primary place field lengths and studying cell pair correlations of the remaining network

Given the increase in spatial information and primary place field lengths in a subgroup of neurons of the sphere-reared rats, we tested whether higher cell-pair correlations in the sphere-reared rats (Fig. 4a, b) were related to lower spatial information or higher place field lengths for place cells during the de novo run on track 1 on Day1. Place cells with the lowest spatial information or highest place field lengths were removed from the analysis of the hippocampal place cell network until sphere-reared rats did not have significantly different spatial information or primary place field lengths compared to cuboid-reared rats, which we tested using the rank-sum test. In this subnetwork of putative pyramidal cells, cell-pair correlations during sleep were subsequently compared across groups.

## Neuronal ensemble similarity score across frame pairs

For frames during sleep, the neuronal ensemble similarity score was computed to detect significantly correlated neuronal ensembles as follows. The total number of spikes per neuron within a frame for all

putative pyramidal neurons were individually counted, and the resulting population-vectors for a pair of frames were correlated with each other using the Pearson's product moment correlation. Subsequently, 500 cell-ID shuffles were constructed for one of the two vectors by randomly permuting cell IDs and the correlation between the same pair of shuffled frames computed to generate shuffled datasets. To derive a neuronal ensemble similarity score, a percentile (expressed from 0 to 1) of the correlation of the real frame pair relative to its respective shuffled distribution was computed. Frame pairs with correlations exceeding the 95th percentile was considered statistically significant. The neuronal ensemble similarity score is a metric of similarity of recruited neuronal ensembles across two frames.

### Clustering of neuronal ensembles across sleep frame pairs

To determine if frames in sleep formed distinct clusters of patterned neuronal activity, the following procedure was adopted. A neuronal ensemble similarity score (n x n) matrix of frame pairs was constructed for each sleep with n frames. The diagonal of the matrix was zeroed. Subsequently, this matrix was transformed into a binary matrix with ones representing significantly correlated frame pairs (>95th percentile with respect to their shuffles) and zeros representing non-significant frame pairs. Next, 100 iterations of a k-means clustering algorithm with the cosine distance used as the dissimilarity metric was applied to each sleep. The number of clusters assigned to the k-means clustering algorithm were set between 2 and 30, based on preliminary exploratory analysis of the data from cuboid-reared rats (control group; we determined that our results were not impacted by the choice of k-means clustering range, on comparing the number of optimal detected clusters in the range 2 to 30 and the range 2 to 60). In each iteration, of the k-means algorithm, 2 to 30 clusters were found in the data and a metric of cluster isolation quality, the silhouette, was computed for each cluster.

$$\text{Silhouette} = \frac{a - w}{argmax(w, a)} \tag{14}$$

where $a$ is the average distance between a cluster and its nearest cluster, while $w$ is the average within cluster distance.

In each iteration, for all the assigned cluster numbers (2 to 30), the average silhouette value was computed. The assigned cluster number (2 to 30) with the highest average silhouette value in most iterations (the mode of the distribution) was selected as the optimal number of clusters in the data. Subsequently, the clustering iteration with the highest average silhouette value (for the modal, optimal number of clusters) was used for further analysis. The binarized neuronal ensemble similarity score matrix was re-sorted based on cluster identity to reveal distinct clusters.

### Clustering of neuronal ensembles across sleep frame pairs for shuffled datasets

In order to determine, the number of clusters in shuffled sleeps, cell IDs were shuffled for the pyramidal cells' activity in frames and the neuronal ensemble similarity score computed for shuffled frame pairs. K-means clustering was performed on the shuffled frames using the same procedure as the real data. For this analysis, the assigned range of cluster numbers was set between 2 to 60 clusters (to limit potential saturation) and the cluster numbers in shuffled datasets of cuboid- and sphere-reared rats were compared with each other.

### Neuronal ensemble similarity of frames with Run

A population vector comprising of peak rate for each pyramidal cell on the track was computed. This vector was correlated with a similar population vector derived for each frame consisting of the total number of spikes for each pyramidal cell per frame. To generate shuffled-datasets, cell IDs were shuffled in the frame 500 times and

correlations with the run population vector were computed for each of the shuffles. A frame with a correlation exceeding the 95th percentile of the shuffled distribution was considered statistically significant.

### Cluster specificity index for individual tracks

To determine if a cluster was likely to be recruited primarily by one of multiple experiences, a cluster specificity index was computed. The cluster specificity index (CSI) was defined as the following:

$$\text{CSI} = \frac{argmax(\sum_{i=1}^{I} Pr_i)}{\sum_{i=1}^{I} Pr_i} \tag{15}$$

where $Pr_i$ is the probability of significant frames for track $I$ for 1 through I tracks explored within a session.

### Pattern separation of two tracks by neuronal ensembles

All the place maps activated on either of the two equal-length tracks explored within a session were stacked. Correlations were obtained for the place maps of neural ensembles for pairs of these tracks as a metric of pattern separation for the two experiences. Solely, to provide an estimate of population vector correlations across a random set of neurons for multiple tracks, we computed chance levels of pattern separation by generating 500 random cell ID shuffles of the place maps and correlating population vectors of the two tracks.

### Correlations of firing rates across tracks

We studied how firing rate properties of neurons across tracks differed between cuboid- and sphere-reared rats. We computed the peak firing rate and mean firing rate of each place cell for each track and running direction and then computed a Pearson's correlation of these two rate population vectors (composed of all place cells) to determine the similarity of rates across tracks or directions.

### Correlations of track corners/ends and middle of tracks on Day2

On Day2, for the contiguous U-shaped 3 tracks, we compared the correlations between neuronal ensembles recruited by track corners/ends and, separately, the track middle. Population vectors for 10 spatial bins (20 cm, 2 cm/bin) centered on opposite track corners or at opposite track ends were correlated (only track corners/ends ~1 m apart were used). Similarly, population vectors for 10 spatial bins centered on the middle of the contiguous tracks (~1 m apart) were correlated to obtain the similarity of neuronal ensembles recruited by middle of different tracks.

### Multiple regression to elucidate the relationship of the individual or ensemble neuronal firing characteristics during Pre-Run sleep or during Run with pattern separation during Run on Day2

We constructed a multiple regression model (the full model) with population vector correlations (continuous variable) across tracks (as a measure of pattern separation) and various predictors. Various predictors from the Pre-Day2Run1 sleep were selected based on the plausibility of a relationship with the future Run while intrinsically representing the sleep activity (and specifically not the relationship of sleep with run). These features were: the average firing rate of pyramidal neurons in Pre-Day2Run1 sleep frames (continuous variable), the average firing rate variation across pyramidal neurons during sleep frames (i.e., variance of average firing rate across neurons, continuous variable), and the average neuronal ensemble similarity across frame pairs (continuous variable). As spatial tuning, quantified as spatial information (continuous variable), increased across development and could potentially impact pattern separation, we identified this as a predictor as well. For reliable model estimation, the model was fit to the cuboid- and sphere-reared rats. To account for other influences of age (continuous variable) or rearing environment (categorical variable), we set these

as predictors as well. A linear relationship was assumed between the variables. To estimate the relative influence of each of these predictor variables on pattern separation (as assessed by population vector correlations across tracks; note that lower population vector correlations indicate higher pattern separation), we removed one predictor at a time and fit a model with the remaining predictors (the reduced models). The influence of that variable was estimated by computing the log-likelihood and then the difference in deviance for each of the two models. We used the likelihood ratio test to determine significant differences between the full and the reduced model.

### Comparison of Days 3–4 with Days 1–2

To determine the role of experience with geometric linearity across days on improvement of spatial representations, various properties of hippocampal neuronal representations during the de novo run on Day1–2 with that of exposures to novel or familiar tracks on Day3–4 were compared. To study p/replay of future Runs on Days3–4, only future run sessions/tracks were used (tracks 1, 4 and 5 or tracks 4 and 6) in sleep prior to exposure to the track.

### Repeated exposure to track 1

Since track 1 (Days1–3) were repeatedly explored over days, we separately tested the effect of repeated experience with geometric linearity on recovery of spatial representations of the same track.

### Index of geometric experience-induced recovery in sphere-reared rats

To estimate the degree of recovery in the sphere-reared rats, we constructed an index of recovery composed of the major parameters we studied across days. These were (1) track-specific preplay, (2) cell assembly plasticity across tracks, (3) number of clusters, (4) similarity of track ends during Run, (5) primary place field length and (6) pattern separation for multiple experiences. The index for each parameter was assigned as the ratio between the average difference after experience with geometric linearity and that before the experience. Thus, a value of 0 indicates full recovery, and 1 indicates no recovery. For each parameter, the first time point at which the parameter could be assessed was used as the baseline (for instance, de novo Day1Run1 for primary place field lengths, and Day2 for pattern separation). For track-specific preplay, cell assembly plasticity and the number of clusters, a value of 0 was assigned on Day4 since complete recovery was observed experimentally. Subsequently, an average for all these parameters (with equal weighting) was computed across these studied parameters to obtain an estimate of the overall recovery, which was plotted on a normalized scale from 0 (sphere) to 1 (cuboid). Additionally, we removed one parameter at a time and recomputed the index to determine the robustness of the estimated recovery.

### Statistical analyses

Parametric statistical tests (ANOVA, one-sample and two-sample Student's t-tests) were used on data that did not violate the normality assumption. Non-parametric statistical tests (Kruskal–Wallis ANOVA, Wilcoxon's rank-sum test or the Wilcoxon signed-rank test) were otherwise employed. Binomial tests were performed against $p = 0.05$ chance to test the significant occurrence of phenomena (i.e., p/replay). Two-sided statistical tests were performed for all comparisons except comparisons against chance/shuffled datasets. Exact or Monte Carlo permutation tests designed by generating Monte Carlo shuffles for permutation of identity (of the group or unit of analysis) were used when appropriate. Comparison between groups with large sample sizes (e.g., million data points) were down-sampled to avoid spurious statistical significance for small effect sizes. A $p$-value threshold of 0.05 ($p < 0.05$) was set for statistical significance. Data were presented in figures as mean ± standard errors of mean unless otherwise stated. Dashed lines depict medians. Key for $p$-values in figures: *$p < 0.05$, **$p < 0.01$, ***$p < 0.005$, ns = not significant.

### Presentation of examples in figures

A moving window (20 ms advanced in steps of 5 ms) was used for displaying theta sequences during locomotion (Fig. 1), and a moving window (20 ms advanced in steps of 10 ms) was used for displaying decoded trajectory sequences in sleep/awake rest (Figs. 2 and 3). Frame duration presented next to each example p/replay frame was rounded up to the nearest 20 ms.

### Reporting summary

Further information on research design is available in the Nature Portfolio Reporting Summary linked to this article.

## Data availability

All data needed to evaluate or extend the conclusions in the article are present in the main article and the Supplementary Figs. Source data are provided with this paper. Raw data used in this study are available from the corresponding author upon request. The very large size of the raw data prohibits their archive on public servers. The raw and processed data used in this article are archived on file servers at the Yale Medical School. No clinical datasets or genetics datasets have been used in this study. Source data are provided with this paper.

## Code availability

The custom codes specific to this study that are needed to interpret, verify, and extend the research in the article can be found[63] at: https://github.com/GDYlab/GDYlabcode and at https://doi.org/10.5281/zenodo.13539650. Additional codes will be available upon request from the corresponding author.

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

## Acknowledgements
We thank K. Loetscher, B. Sanders and J. Sibille for help with data collection and analysis, and all the members of the Dragoi lab for comments and help. This work was supported by NIH grants R01NS10491, R01MH121372, and R35NS132342 to G.D.

## Author contributions
U.F. and G.D. collected and analyzed the data. G.D. conceived and designed the study. G.D. and U.F. wrote the manuscript.

## Competing interests
The authors declare no competing interests.
