## [Peer Review File · Nature Communications]

Experience of Euclidean geometry sculpts the development and dynamics of rodent hippocampal sequential cell assembliesREVIEWER COMMENTS

Reviewer #1 (Remarks to the Author):

I have read and evaluated the manuscript titled “Experience of Geometry Sculpt the Development and Dynamics of Hippocampal Sequential Cell Assemblies.” This exceptional scientific study examines the developmental (preconfigured) and learned (configured) representations of a linear dimension by hippocampal neuronal ensembles. The experimental design is well developed and builds on a solid foundation of evidence that much of the coding in the neuronal assemblies in the hippocampus uses pre-existing sequential firing patterns to represent and learn about space. This novel study includes critical controls for behavioral, endocrine (stress) and physiological confounds. The writing and figures are of the highest quality. The authors conclude that being reared in an Euclidean environment benefits the neuronal coding for such spaces.

I am incredibly enthusiastic about this work. It will have a broad and lasting impact in several fields, from neurobiology to psychology (and philosophy). I have listed several minor conceptual questions below about how the authors interpret their findings, as I have no technical concerns.

Please explain the different notions captured by the phrases “driven by reduced preconfigured network repertoires” and “enriches the hippocampal repertoire.” These seem to be a bit relativistic. Can the data speak to whether pre-configurations were reduced in sphere-reared animals or enhanced in cubic-reared animals?

If linear configurations are early-life experience dependent, how can they be strictly preconfigured? The logic seems circular. Instead, why not refer to this as an early configuration during a critical learning period in development?

On Page 21, Line 10 is “p/re-configured” meant to convey the authors’ ambivalence about the role learning has on the neuronal sequences that emerge from development (preconfigured) rather than experience (reconfigured)? It is not always clear as different forms of similar terms appear throughout the text (preconfigured, pre-configured, p/re-configured, etc.).

Do the authors mean that linear dimensionality is preconfigured in sequences, and not conceptual space? This should be clarified. Sometimes it is not clear in the text which of these distinct meanings are intended.

Are cubic shapes ecologically valid since squares do not naturally occur in the environment generally (not with any regularity)? As such, why would cubic (unnatural) experiences actually improve representational orthogonalization and plasticity? This effect may more simply be due to more environmental features and dimensions rather than a specific geometry.

Is it possible or probable that the hippocampal ensemble benefits from cubic rearing is specific to the linear track? Would sphere-reared rats represent and distinguish similar circular tracks better? Is this a prediction of the observed blurring of starts and stops in sphere-reared animals?

Why is additional training considered recovery (Figure 7) following multiple days of experience rather than de novo learning?

Lastly, these scientific questions are complex, and definitive answers are hard to come by—I commend this effort. In that spirit, I invite the authors to speculate about other experimental rearing environments that might illuminate this issue going forward. For example, the design is Euclidean deprivation. Maybe future studies with different fractal environments that are closer to nature, but still mathematically controllable, would be useful?

Reviewer #2 (Remarks to the Author):

In this manuscript, Farooq and Dragoi investigate the influence of early life experience in the development of hippocampal representations of space and intrinsic organization of cell assembly dynamics. They set to ask the question of whether the hippocampal internal cognitive model of space is largely innate (i.e., pre-configured from birth due to a genetically encoded developmental program) or if instead it mostly emerges due to experience with Euclidean spaces (such as typical vivarium cubic cages). To tackle this question, they took the original approach of rearing rats from birth in spherical cages that therefore lack basic linear spatial features. They then expose regular and sphere-reared rats to linear tracks while recording hippocampal ensemble activity. They found that this distinct early experience worsened spatial tuning of hippocampal place cells, the neural discrimination of multiple linear environments and the richness of pre-existing assembly patterns in the hippocampus. However, they found that the characteristic hippocampal time-compressed sequences representing spatial trajectories were still present and undergo learning-related plasticity. Experimental and analytical methods employed in this study are sound and rigorous. The results presented support the authors conclusions and add a novel and important perspective to a fundamental question in the field. I have a few comments that need to be addressed to improve the accuracy and clarity of some analysis. I believe this work will be an important addition to field and of broad interest for the community.

1. The authors need to better explain why a transparent home cage was used for the cuboid-reared group, but a white translucent home cage was used for the sphere-reared group, because a transparent home cage can give the cuboid-reared group better sensory enrichment (e.g., better visual access to the outside environment), which can be a confound for the current experiment (i.e., geometric linearity is not the only difference that the two groups had during development).

2. The authors need to explain why a sliding window (20 ms shifted by 5 ms) is used for theta sequence quantification, but a non-overlapping window is used for replay quantification. The authors concluded that the two groups showed similar theta sequence compression (page 7), but geometric linearity “attenuated the incidence of time-compressed linear p/replay” (page 8). It is possible that the sliding window can overshadow the discontinuity of theta sequences in the sphere-reared group. Therefore, it is better to use the same detection and statistical methods for both p/replay and theta sequences for a fair comparison of these two types of sequences. In addition, the author reported a difference in the proportion of significant replay sequences between the two groups (Fig. 2c). What are the proportions of significant theta sequences of the two groups (Fig. 1g only showed the quadrant ratio)?

3. To better quantify the representational warping in replay sequences, instead of removing the start and end time bins of the candidate replay events (because replay can start from any locations, not just from the reward wells; also removing the bins can reduce the statistic power for subsequent shuffling procedures), the authors can remove the start and end from the rate maps of each cells (i.e., place-field templates for replay detection) and re-detect the replay sequences. Similarly, the same method can be used for theta sequence detection for comparison (related to the 2nd comment).

4. Related to the previous point, if sphere-reared group indeed represented the linear space as a circular environment as the authors suggested (shown in their schematic, Fig. 3e), using circular tests, instead of linear regressions, for place fields and replay sequence quantification presumably will yield better detection for the sphere-reared group than the cuboid-reared group. This measure can strengthen the conclusion of the paper.

5. To rule out the contribution of the difference in spatial information for replay detection, the authors dropped cells with low spatial information in sphere-reared group until spatial information is matched across both groups. However, dropping cells will also yield lower cell counts, which will result in lower detection quality for the sphere-reared group. Therefore, a better way is to subsample both groups to match the cell number, as well as spatial information, across groups.

Minor:

1. For the cluster analysis for sleep frames, the authors need to explain why for the assigned cluster number 2 to 30 (Methods, page 53), the shuffled data had number of clusters above 40 (Fig. S5e). Was a different cluster range used for shuffled data, and why? If the same cluster range is used for shuffled and real data, is the number of clusters for the cuboid-reared group significantly lower than that of their shuffled data?

2. I believe that labels of Fig. S6g are wrong. The labels of the axes don't make sense, and the legends of the cuboid and sphere groups on the bottom left panel seems swapped. Please correct this figure.

3. In Fig. S7g, can the authors provide some explanations why the decreasing trend for the asymmetric place representation disappeared for the cuboid-reared group for Day3Run1? This is an important measure used in many figures. However, this result raised the question of how reliable this measure is.

4. To compare the pattern separation of two tracks, instead of randomly shuffling cell IDs (Methods, page 54), a better way could be swapping the rate maps of the same cells across two tracks (Tirole et al., 2022).

5. In Fig. S3f (bottom right panels), can the authors provide some explanations why the stripes for the decoded locations were not located at the end of the track (0 or 1), which seems to be against with their other measures of representational warping. In addition, similar stripes seem appear for the cuboid-reared group as well.

6. Fig. 4f, "number of clusters during Day1 sleep sessions", should it be "number of clusters during Pre-Day1Run1 sleep sessions"?

7. Can the author provide some explanations for the difference in number of clusters shown in Fig. 6b middle panel (not significant) versus in Fig. 4f (significantly different)?

8. Page 5, "~ 90 min" should be "~ 90 mins".

Antonio Fernandez-Ruiz

Reviewer #3 (Remarks to the Author):

In this manuscript, Farooq and Dragoi examined differences between rats that were reared from birth in spherical and in cuboid environments in terms of CA1 spatiotemporal representations during both wake and sleep. Similar to cuboid-reared rats, sphere-reared rats exhibited place cell activity and theta sequences during navigation, as well as place cell sequence preplay, replay and the enhancement of place cell sequence replay by experience, during sleep. However, neural codes relatively lacked complexity/specificity in sphere-reared rats, with cells showing more similar encoding of different linear tracks and of the two halves of a linear track, as well as a poorer repertoire of preconfigured neuronal patterns during sleep. These differences faded away following four days of training in the linear tracks.

This is a comprehensive study that addresses various facets of CA1 spatiotemporal encodings during wake and sleep for rats reared in spherical vs cuboid environments. I have a few comments related to the presentation of results and some potentially lacking elements of discussion.

_ In general, I found the result section not always easy to read, partly due to a lack of concision and a non-straightforward organization of analyses.

For instance, an interesting finding is that sphere-reared rats showed symmetrical encoding of the two halves of a linear track. In my view, the presentation of the results related to this point could be more straightforward.

Several population analyses with some degree of redundancy are first presented. Not only understanding each of these analyses requires some effort, but it is also not straightforward to figure out their combined meanings. With some thinking (largely based on Supplementary Fig. 3 and Supplementary Fig. 4), one can infer that some of the cells in sphere-reared rats should have the same 2 firing fields, equidistant to track ends, for both travel directions. This is only partially mentioned in the text, in the last sentence of the 3rd paragraph of the section.

I think the section would be easier to read if the authors first showed the rate maps of a few cell/symmetric cell examples, each running direction shown separately, and the respective proportion of various cell representations (similar to Suppl Fig 4c), and then presented the population analyses which are relatively simple to interpret from a higher proportion of symmetric cells in sphere-reared than cuboid-reared rats.

_ In several instances, the authors put forward the idea of a representational 'warping' of linear space in sphere-reared rats. While it is not an inaccurate description of the results in Fig 3c and d, it might be somewhat misleading conceptually as the reward/geometrical symmetry of the two halves of the track might likely be the critical factor of the apparent warping (representation might not be 'warped' in a linear track without rewards/endings). As is, some readers might be misled into thinking that rearing in spheric environments generally bias the network to generate warped representations of linear space.

_ Potential mechanisms for the symmetric encoding of track ends and for the larger proportion of such cells in sphere-reared rats is obscure or lacking in the discussion. Cells symmetrically encoding track ends are somewhat reminiscent of landmark/boundary/object-vector cells (Lever et al. 2009, Deshmukh & Knierim 2013, Geiller et al. 2017, Hoydal et al. 2019), which tend to fade away with learning of environments through competitive learning (Kim et al. 2020).

We thank the 3 reviewers for evaluating our manuscript and for their constructive, positive feedback, which has substantially improved our manuscript. The revised manuscript includes 1) improved text with new, concise additional sentences providing further explanation or regarding interpretation following reviewers' feedback, 2) new analyses and figure panels to support our findings based on the reviewers' recommendations, 3) an improved presentation of figure panels and methods to conform with the journal guidelines. In the revised manuscript Main text and figure captions, the sentences addressing the reviewers' points and the changes made based on reviewers' recommendations are in blue color. Please find below our detailed comments in response to each point the reviewers raised in italics.

REVIEWER COMMENTS

Reviewer #1 (Remarks to the Author):

I have read and evaluated the manuscript titled "Experience of Geometry Sculpt the Development and Dynamics of Hippocampal Sequential Cell Assemblies." This exceptional scientific study examines the developmental (preconfigured) and learned (configured) representations of a linear dimension by hippocampal neuronal ensembles. The experimental design is well developed and builds on a solid foundation of evidence that much of the coding in the neuronal assemblies in the hippocampus uses pre-existing sequential firing patterns to represent and learn about space. This novel study includes critical controls for behavioral, endocrine (stress) and physiological confounds. The writing and figures are of the highest quality. The authors conclude that being reared in an Euclidean environment benefits the neuronal coding for such spaces.

I am incredibly enthusiastic about this work. It will have a broad and lasting impact in several fields, from neurobiology to psychology (and philosophy). I have listed several minor conceptual questions below about how the authors interpret their findings, as I have no technical concerns.

Please explain the different notions captured by the phrases "driven by reduced preconfigured network repertoires" and "enriches the hippocampal repertoire." These seem to be a bit relativistic. Can the data speak to whether pre-configurations were reduced in sphere-reared animals or enhanced in cubic-reared animals?

We thank the reviewer for their enthusiastic review and constructive feedback, which improved our manuscript. We agree with the reviewer that because our results show the experience-dependence of certain aspects of hippocampal neurodevelopment, whether spheres reduce, or cuboids increase/enrich network configuration is primarily relative to the other. Since we show that spatial experience plays specific roles as animals experience space when they move in different geometric contexts, we can safely conclude that a 'change' in the nature of experience across these two rearing environments is associated with a change in hippocampal network repertoire. Whether the default repertoire is the one observed after sphere-rearing, which is merely enriched via experience with Euclidean geometry or, conversely, the default repertoire is the one observed after cuboid-rearing, which is merely reduced by un-natural deprivation via sphere-rearing, remains to be determined. In either case, we showed that emergence of a

preconfigured hippocampal repertoire (envisioned as a network competence) does not depend on early-life experience with Euclidean geometry despite the fact that experience with the latter does augment the size and complexity of that repertoire (envisioned as a network performance). We replaced the term 'reduced' with 'relatively reduced' and 'reduction' to 'relative reduction'. We reserved the terms 'reduced' or 'increased/enriched' for measuring the impact that exposure to geometric linearity after sphere-rearing had on properties of hippocampal neurons and ensembles.

In Fig. 6e, f, we show that the intrinsic repertoire (via frame pair similarity) has a relationship with future pattern separation when other variables are accounted for. We summarized this in the abstract as follows: "sphere-rearing [...] and impaired neuronal pattern separation and plasticity for multiple linear tracks, partly driven by reduced preconfigured network repertoires." In line with the reviewer's recommendation, we have now replaced "partly driven by reduced" with "coupled with relatively reduced" preconfigured sleep network repertoires.

The total possible network configurations in sleep (or the sleep network space) that can be recruited to represent future experiences form the hippocampal repertoire. We use the phrase 'enriches the hippocampal repertoire' to indicate a relative increase in/induction of more complexity in the intrinsic/sleep repertoire of preplay/preconfigured dynamics. Based on the reviewer's recommendation, we added the term 'sleep' to the first mention of the word repertoire in the abstract to clarify that experience with Euclidean geometry enriches the hippocampal sleep repertoire. Additionally, at various points in the manuscript, we ensured that our statement on enrichment of the repertoire was qualified as follows: Euclidean geometry/geometric linearity enriches the hippocampal repertoire (since such experience was unavailable to sphere-reared rats).

We modified the manuscript to clarify these phrases and the point on relative configuration as recommended by the reviewer.

If linear configurations are early-life experience dependent, how can they be strictly preconfigured? The logic seems circular. Instead, why not refer to this as an early configuration during a critical learning period in development?

The dichotomy between network competence and network performance in expressing preplay and replay (also discussed in the above point in the context of network repertoire) helps explaining the point raised by the reviewer. Particularly, network competence to generally express preplay of future novel trajectories (i.e., network preconfiguration) emerges in absence of experience with Euclidean geometry and is expressed for one track on Day1. However, early-life experience with Euclidean geometry (vs. lack of such experience) relatively increases network performance to express a larger repertoire of distinct preplay for distinct future trajectories, as shown on Day2 for 3 tracks in cuboid- vs. sphere-reared rats. Based on our results, we argue that the early-life 'emergence' of neuronal sequences in the hippocampus for a single linear track is Euclidean geometry experience-independent or preconfigured (i.e., significant preplay is present in sphere-reared rats), while experience of Euclidean geometry further configures the network such that the proportion of such distinct sequences (their repertoire) is increased. Therefore, inferred intrinsic developmental programs and explicit geometric experience might play 'different' and not completely overlapping roles in the

expression of more complex repertoire in the cuboid-reared group. Additionally, it is important to note here that the elements of the network discussed above that are configured in early life remain 'preconfigured' (i.e., configured before) relative to their adult usage for representing future spatial experiences. Whether an early critical/preferential learning period exists during development after which exposure of sphere-reared rats to Euclidean geometry does not increase their network performance/repertoire is a very attractive idea. However, since we do not have data to currently support or refute this idea, we suggest this as a potential next experiment for the future. We modified the manuscript to clarify all these points as recommended by the reviewer.

On Page 21, Line 10 is "p/re-configured" meant to convey the authors' ambivalence about the role learning has on the neuronal sequences that emerge from development (preconfigured) rather than experience (reconfigured)? It is not always clear as different forms of similar terms appear throughout the text (preconfigured, pre-configured, p/re-configured, etc.).

We removed the term p/re-configuration from this sentence and the rest of the manuscript as we agree with the reviewer that this could lead to potential misunderstanding. We generally use the term p/replay for preplay and replay when referring to approaches and findings that refer to and characterize both processes. Since we show that adult network 'preconfiguration' (sleep network activity which is utilized by future experiences) is composed of an element of early-life geometric experience as well, we thought this justifies the term p/reconfigured as well, specifically when referring to preplay. While early-life experience plays a role in configuring the network, the sleep activity patterns remain 'pre-configured' (configured before) a future spatial experience they are recruited by. We modified the manuscript to define and improve consistency of these terms across the revised manuscript and clarified the term preconfiguration as recommended by the reviewer.

Do the authors mean that linear dimensionality is preconfigured in sequences, and not conceptual space? This should be clarified. Sometimes it is not clear in the text which of these distinct meanings are intended.

Linear dimensionality has been tested and shown to be preconfigured in sequences in adult and developing animals (Dragoi and Tonegawa, 2011, Nature; Dragoi and Tonegawa, 2013, PNAS; Grosmark and Buzsaki, 2016, Science; Farooq et al., 2019, Neuron; Farooq and Dragoi, 2019, Science). At the same time, it has been proposed that the hippocampus represents conceptual spaces in the form of cognitive maps, which might use the same neural architecture (e.g., Bellmund et al., 2018, Science). Here, we showed that even in the absence of experience with geometric linearity, sequential activation of hippocampal neurons representing linear space still emerges. This could indeed be interpreted as consistent with the existence of an innate, generic concept of space. Moreover, here, we demonstrate that experience with linear dimensionality is required for neurodevelopment of an increased repertoire of these sequences that could enable rapid generation of distinct neural representations for multiple linear environments, rather than a generic representation of linear space. Therefore, while our experiments explore the impact of linear dimensionality on neuronal sequence development, these could extend to representation of conceptual spaces as well. We clarified this point in the revised manuscript discussion as recommended by the reviewer.

Are cubic shapes ecologically valid since squares do not naturally occur in the environment

generally (not with any regularity)? As such, why would cubic (unnatural) experiences actually improve representational orthogonalization and plasticity? This effect may more simply be due to more environmental features and dimensions rather than a specific geometry.

As we proposed in Figure 1a, cuboid environments indeed could 'train' the hippocampus via direct experience with a number of spatial features: linearity, linear boundaries, planarity, corners (which also increase dimensionality of the environment), vertical walls, distinct beginning and end of trajectories, potential landmarks. Conversely, spherical environments could 'train' the hippocampus via direct experience with curvature. While cubic shapes might not occur frequently in the wild habitat of rodents, spatial features of cubic shapes are ubiquitous in Euclidean world where rodents (and humans) live. Therefore, these spatial features could further configure the hippocampal representation of space via their direct experience in addition to the inferred innate concepts of space. For instance, orthogonalization of behavior across 90-degree corners might drive network orthogonalization and pattern separation, and once sufficient pattern separation is achieved, this might enable plasticity of those distinctly represented experiences. Please note that these spatial features (like 90-degree corners also increase the dimensionality of the environment as pointed out by the reviewer. In this regard, we agree with the reviewer that it is the role of multiple features of the environment that we are investigating. We clarified this in the revised manuscript and suggested studying impact of individual spatial features across Euclidean and non-Euclidean environments is a potential future direction in the revised manuscript as recommended by the reviewer.

Is it possible or probable that the hippocampal ensemble benefits from cubic rearing is specific to the linear track? Would sphere-reared rats represent and distinguish similar circular tracks better? Is this a prediction of the observed blurring of starts and stops in sphere-reared animals?

The current evidence we provide indicates a higher degree of representational warping in the sphere-reared rats that is observed at the level of network manifold, in a more abstract manner than simple curvature. This effect is contributed by a lack of pattern separation between the 2 ends of a linear track, which appears warped at the manifold level. Additionally, our new results show that usage of circular statistics (under the assumption that the linear space is circular) instead of linear statistics for quantifying p/replay diminishes the differences across the two groups in trajectory sequences, specifically replay (revised Supplementary Fig. 4h). On the other hand, having experience with only curved trajectories instead of geometric linearity might provide 'insufficient' training to the hippocampal network to represent multiple features of spatial information, including segmentation, abrupt changes in orientation, corners, linear boundaries, etc., which might extend to any type of geometry (Euclidean or curved). Therefore, our current data is consistent with both possibilities and further work will be required to determine this. We added this to the discussion of our revised manuscript as recommended by the reviewer.

Why is additional training considered recovery (Figure 7) following multiple days of experience rather than de novo learning?

We agree that de novo learning could contribute to recovery of underdeveloped spatial representation features in sphere-reared rats, which may be considered the default features of normally-reared rats. In this respect, the differences observed between sphere- and cuboid-

reared rats could be envisioned as deficits of the former driven by spatial deprivation, and their emergence after training as recovery and not only de novo learning. Indeed, after de novo learning and overnight sleep we observe such improvements, including improvement in sleep clusters (revised Fig. 6b). However, overall, substantial recovery requires several days of repeated experience (revised Fig. 7 and 8). We clarified this point in the revised manuscript as recommended by the reviewer.

Lastly, these scientific questions are complex, and definitive answers are hard to come by—I commend this effort. In that spirit, I invite the authors to speculate about other experimental rearing environments that might illuminate this issue going forward. For example, the design is Euclidean deprivation. Maybe future studies with different fractal environments that are closer to nature, but still mathematically controllable, would be useful?

In the revised manuscript, we have shared our hypotheses and future directions that could shed further light on the role of geometry in hippocampal neurodevelopment, which is about parsing out the role of our identified spatial features within Euclidean environments listed in Fig. 1a. We thank the reviewer for providing us with the opportunity to further elaborate on these topics, in which we have attempted to remain concise, in line with the recommendation of reviewer 3 and the guidelines provided by Nature Communications.

Reviewer #2 (Remarks to the Author):

In this manuscript, Farooq and Dragoi investigate the influence of early life experience in the development of hippocampal representations of space and intrinsic organization of cell assembly dynamics. They set to ask the question of whether the hippocampal internal cognitive model of space is largely innate (i.e., pre-configured from birth due to a genetically encoded developmental program) or if instead it mostly emerges due to experience with Euclidean spaces (such as typical vivarium cubic cages). To tackle this question, they took the original approach of rearing rats from birth in spherical cages that therefore lack basic linear spatial features. They then expose regular and sphere-reared rats to linear tracks while recording hippocampal ensemble activity. They found that this distinct early experience worsened spatial tuning of hippocampal place cells, the neural discrimination of multiple linear environments and the richness of pre-existing assembly patterns in the hippocampus. However, they found that the characteristic hippocampal time-compressed sequences representing spatial trajectories were still present and undergo learning-related plasticity. Experimental and analytical methods employed in this study are sound and rigorous. The results presented support the authors conclusions and add a novel and important perspective to a fundamental question in the field. I have a few comments that need to be addressed to improve the accuracy and clarity of some analysis. I believe this work will be an important addition to field and of broad interest for the community.

1. The authors need to better explain why a transparent home cage was used for the cuboid-reared group, but a white translucent home cage was used for the sphere-reared group, because a transparent home cage can give the cuboid-reared group better sensory enrichment (e.g., better visual access to the outside environment), which can be a confound for the current

experiment (i.e., geometric linearity is not the only difference that the two groups had during development).

We thank the reviewer for their review and constructive feedback which improved the manuscript. Prior work has suggested that there is no significant impact of vision itself on the development of place cells (tested in blind rats, Save et al., 1998, Journal of Neuroscience), head-direction cells (tested in genetically blind mice, Asumbisa et al., 2022, Nature Communications), and spatial signals (earlier-developing head direction cells as well as grid cells, tested in a frosted cage versus a cage with no walls allowing clear visual access to the room unlike walled cages) in the upstream medial entorhinal cortex (Ulsaker-Janke et al., 2023, PNAS). These relevant papers are cited in our discussion section. In our experiment, the cuboidal home cages used for cuboid-reared rats had amber color transparent walls and were placed on a table without walls. Extra-cage relevant cues were not available within 1.5 m in the controlled environment for the rats throughout rearing, except the room vertical walls and solid black curtain for cutting off the remaining of the room (pregnant females or P4 rats – 10 days before eye-opening – were brought to the laboratory long ahead of eye opening day). In order to ensure that no physical and visual experience with geometric linearity was being provided to the sphere-reared rats, we used a translucent spherical homecage, which allowed light to enter but limited physical or visual access to geometric linearity. For both groups, the experimenters themselves provided access to food, water, light, cage cleaning during development to ensure similar provision of these across groups. Despite lacking both visual and physical access to geometric linearity, p/replay for a linear track emerged during neurodevelopment in the sphere-reared rats. We improved the explanation of these critical experimental details in the revised Methods and in the Main text, as recommended by the reviewer.

2. The authors need to explain why a sliding window (20 ms shifted by 5 ms) is used for theta sequence quantification, but a non-overlapping window is used for replay quantification. The authors concluded that the two groups showed similar theta sequence compression (page 7), but geometric linearity “attenuated the incidence of time-compressed linear p/replay” (page 8). It is possible that the sliding window can overshadow the discontinuity of theta sequences in the sphere-reared group. Therefore, it is better to use the same detection and statistical methods for both p/replay and theta sequences for a fair comparison of these two types of sequences. In addition, the author reported a difference in the proportion of significant replay sequences between the two groups (Fig. 2c). What are the proportions of significant theta sequences of the two groups (Fig. 1g only showed the quadrant ratio)?

We agree with the reviewer that the choice of analyses methods might impact detection of sequences (i.e., usage of a sliding window, quadrant ratio vs. weighted correlation, and average versus individual significant frame proportions). We used a sliding window to increase resolution within theta cycles, potentially revealing any differences across the two groups at a finer timescale than the binning window (despite which we observed no significant difference between the two groups). Additionally, we reasoned that while the weighted correlation is an important measure for sequence analysis, the advantage of using the quadrant ratio for theta sequences is to compute and analyze the probabilities around/centered on the current location of the rat, an important property of theta sequences (i.e., theta sequences bind the immediate past-present-future), which motivated our choice of this measure for theta sequence analysis. Further, we reasoned that as more cells are likely to be coactivated in ripples versus theta

cycles, and theta sequences are less variable compared to ripple sequences, a more reliable estimate for theta sequence would emerge on average theta sequence quantification for the whole run session for one direction. We acknowledge that the reviewer's recommended analyses are indeed important analyses and will strengthen our conclusions.

Therefore, we have now conducted the following new analyses to address the reviewer's points:

1) We computed the quadrant ratio using a non-overlapping 20 ms time windows for theta sequences. We report that theta sequences using this method are present in cuboid-reared and sphere-reared rats and are not significantly different across groups, in agreement with our original conclusion (in our original Fig. 1e). These results are now presented in revised Fig. 1e based on the reviewer's recommendation (while the theta sequence quadrant ratio results with an overlapping time bin window are presented in revised Supplementary Fig. 1k). We additionally tested if the track ends were impacting our theta sequence analysis, by progressively removing place maps from track ends and studying the theta sequence quadrant ratio across groups (using a non-overlapping 20 ms time window). We found no significant impact of removing up to 30 cm from track ends (presented in revised Supplementary Fig. 1l).

2) We computed the proportion of individual theta cycle frames with significant quadrant ratio (for cycle frames with ≥ 5 active cells to be consistent with trajectory p/replay analysis) compared to a within frame time-bin shuffle (using a 20 ms non-overlapping temporal window). The proportion of such frames, equivalent with the proportions of significant theta sequences requested by the reviewer, was similar across the cuboid- and sphere-reared rats. We present these results in revised Supplementary Fig. 1m.

3) Using the same methodology and statistics as those used for p/replay sequences, we computed weighted correlations for cycles with ≥ 5 active cells and computed the proportion of significant forward going theta sequences compared with time-binned shuffles (using a non-overlapping 20 ms temporal window). Since theta sequences are forward-going, we report that the proportion of significant forward going theta sequences (at $p < 0.05$) are similar across the two groups. We present these results in revised Supplementary Fig. 1n. We used ($p < 0.05$ for forward going theta sequences alone) since using this procedure we do not observe significant proportions of negative going theta sequences (at $p < 0.05$, data not shown, as this is not central to our main findings). However, additional lower p-value thresholds did not impact our results (similar theta sequences in cuboid- and sphere-reared rats). Thus, we conclude that our observed results on theta sequences vs. p/replay are robust to differences of measures/methodology.

3. To better quantify the representational warping in replay sequences, instead of removing the start and end time bins of the candidate replay events (because replay can start from any locations, not just from the reward wells; also removing the bins can reduce the statistic power for subsequent shuffling procedures), the authors can remove the start and end from the rate maps of each cells (i.e., place-field templates for replay detection) and re-detect the replay sequences. Similarly, the same method can be used for theta sequence detection for comparison (related to the 2nd comment).

We agree with the reviewer that p/replay can start from any location of the track. Consequently, we hypothesized that representational warping of a sequence could occur no matter where in space a sequence starts or ends, as this phenomenon extends beyond the track ends during

the run (presented in revised Fig. 3c) and might do so in the sleep as well. For this reason, we argued that removing the start or end of frames in time might be more informative, than removing the start or end locations in space on the track. However, since both during run and sleep (Fig. 3) we observed the differences between the two groups were most marked between/near the track ends (as these locations are furthest apart in space), we finally used this conservative interpretation for our analysis on trajectory sequences. We have now added new further explanation to the revised methods to clarify this point. For this analysis, our approach was cumulative, where sequence frames already significant for the experience were kept 'as is', while the non-significant frame beginnings or ends in time were removed similarly in cuboid and sphere-reared rats. To ensure sufficient statistical power, we only considered frames with ≥ 5 remaining bins for further analysis. The total cumulative significant sequences in both groups were compared after removal of beginnings or ends. The same procedure was used for both groups. To test for any non-specific impact of this procedure, we also followed the same procedure on time bins in the frame middle, i.e., we progressively removed time bins from the frame middle and studied the differences in p/replay sequences across cuboid- and sphere-reared rats. Unlike removal of time bins from frame starts/ends, after removal of time bins from the frame, middle p/replay did not abolish the difference in trajectory replay (revised Supplementary Fig. 4c).

As elaborated by the reviewer and above, removing specifically place maps from track ends before p/replay quantification and detection of frames, tests the hypothesis that there is a track end swapping/confusion within these sequences, which is an important test for the impact of sphere-rearing on depiction of track ends. In line with reviewer's recommendation, we now conducted new analyses by removing the two track ends (0 to 30 cm from track ends in increments of 5 cm) from the rate maps of each cell before detecting frames and performing Bayesian decoding. The same procedure was used for both cuboid and sphere-reared rats. We next tested if this procedure reduced the difference between cuboid- and sphere- p/replay. We found that this procedure indeed significantly reduced the difference in p/replay sequences across cuboid- and sphere-reared groups (presented in revised Supplementary Fig. 4f), which was specific to the track end, since applying a similar procedure to the track middle had no significant impact on the differences in p/replay across groups (revised Supplementary Fig. 4g). Although we did not observe any significant difference in theta sequences across groups for the non-reduced data (see revised Fig. 1e and Supplementary Fig. 1 k-n), we also applied a similar procedure to the average quadrant ratio of theta sequences (which was chosen due to the reasons listed in detail in response to point 2 (i.e., its ability to compare probabilities around the current location on the animal and provide a reliable estimate of the whole session), and a non-overlapping 20 ms temporal window as recommended by the reviewer. This procedure did not impact theta sequences in any of the two groups (in revised Supplementary Fig. 1l, presented with other theta sequence results). We additionally tested this hypothesis on pooled proportions of significant theta frames across all animals and also found no significant impact on the differences in proportion of significant theta frames across groups with the weighted correlation and a non-overlapping time bin window, $p > 0.05$, permutation tests, data not shown). Together, we used 4 different analytical methods (each with differing assumptions and studying to some degree non-overlapping aspects of this phenomenon): (1) cutting frame starts/ends in time and quantifying cumulative proportion p/replay across groups, 2) performing a T-1 contingency analysis on all sleep frames to uncover short-timescale spatio-temporal relationships 3) removing place maps at track ends and re-computing p/replay and 4) using linear vs. circular

statistics (next point) for p/replay quantification. Using these methods, we now show that sleep decoded spatio-temporal activity dynamics (like the Run) also exhibit hints of representational warping.

4. Related to the previous point, if sphere-reared group indeed represented the linear space as a circular environment as the authors suggested (shown in their schematic, Fig. 3e), using circular tests, instead of linear regressions, for place fields and replay sequence quantification presumably will yield better detection for the sphere-reared group than the cuboid-reared group. This measure can strengthen the conclusion of the paper.

One crucial piece of evidence for representational warping is the network manifold (revised Fig. 3b-e), which is contributed by a pattern separation between the 2 ends of the linear track. We first studied linear weighted correlations of decoded space with time in cuboid- and sphere-reared rats to study the role of early-life experience of 'linearity' on representing 'linear' spaces by these sequences. While linear trajectory sequences were lower in sphere-reared rats, we observed a higher similarity of track ends (and other symmetric locations), which is suggestive of a representational warping of linearity. Thus, while circular statistics for p/replay (as in Grosmark et al., 2021, Nature Neuroscience) would likely aid in revealing sequences spanning the whole track (and exhibiting a swapping/confusion at the ends), it will likely not account for warping in sequences starting and ending at other locations (as indicated in the response to point 3), and thus precluded our initial use of these measures.

In line with the reviewer's recommendation, we have now also used the circular-linear weighted correlation to quantify p/replay sequences in the revised manuscript (revised Supplementary Fig. 4h). We assumed space is circular rather than linear for this calculation and used the same metric for the within frame time-bin shuffles. Interestingly, the difference between sphere and cuboid-reared rats using circular statistics specifically in trajectory replay was lower compared to linear weighted correlations, that was not the case for preplay ($p=0.07$ tending to significance, permutation test), likely contributed by a higher tendency to start at other locations or because of how preplay is organized (i.e., more likely to be linear). Additionally, we note that since preplay preferentially has lower frame duration, it might be not as amenable to this analysis as to replay, which initially motivated our T-1 contingency analysis for revealing short-timescale spatio-temporal relationships within frames amidst various track locations (Fig. 3g, right). As explained above, this method will primarily isolate 'sequences' exhibiting a swapping of track ends and not as much a representational warping of other locations on the track. We did not find any clear existing equivalent of this analysis for individual place maps. Thus, due to the reasons discussed in this and previous points on quantifying representational warping, to quantify the instantiation of this phenomenon at the individual place map level, our preferred metric was the 'symmetry index' for place maps (which allowed us to capture the symmetry of the response). Using this method, a firing-rate index for symmetric locations on the tracks was computed and compared across groups, and we found a small, but significantly higher proportion of symmetric neurons in the sphere-reared rats (revised Fig. 4f; on recommendation of Reviewer 3). Together, using 4 methods (summarized at the end of point 3), we show a degree of representational warping in the sphere-reared rats.

5. To rule out the contribution of the difference in spatial information for replay detection, the authors dropped cells with low spatial information in sphere-reared group until spatial information is matched across both groups. However, dropping cells will also yield lower cell

counts, which will result in lower detection quality for the sphere-reared group. Therefore, a better way is to subsample both groups to match the cell number, as well as spatial information, across groups.

To ensure matching of spatial information as well as cell numbers in line with reviewer's recommendation, we now dropped cells from both the top spatial information ranks from the cuboid-reared rats and the bottom spatial information ranks from the sphere-reared rats until spatial information across two groups. Following this new procedure, we observed significant p/replay in cuboid- and sphere-reared rats, and significantly higher p/replay in cuboid-reared rats compared to sphere-reared rats. These results indicate that single cell spatial information alone does not solely account for the differences in trajectory p/replay across the two groups. These results are now presented in the revised Supplementary Fig. 2f. While the results are qualitatively similar with our previous ones, we present both sets of results since removing only the lowest spatial information cells in the sphere-reared rats until spatial information is matched across groups might provide additional information for the reader.

Minor:

1. For the cluster analysis for sleep frames, the authors need to explain why for the assigned cluster number 2 to 30 (Methods, page 53), the shuffled data had number of clusters above 40 (Fig. S5e). Was a different cluster range used for shuffled data, and why? If the same cluster range is used for shuffled and real data, is the number of clusters for the cuboid-reared group significantly lower than that of their shuffled data?

For statistical comparisons of our data, we now directly compared the number of clusters in Pre-Day1Run1 sleep data and shuffled datasets across groups using clustering that was done on 2 to 60 clusters. We added more details to the revised methods section elaborating on the procedure for the data and shuffled dataset for this analysis, as recommended by the reviewer. Increasing the range of clusters for the cuboid and sphere-reared rats did not impact the number of detected clusters in any of the sleep sessions (revised Supplementary Fig. 5e). We observe that both groups have significantly lower number of clusters in the real data compared to the shuffled datasets indicating a higher degree of structure in data vs. shuffle. We now present these results in revised Supplementary Fig. 5g.

2. I believe that labels of Fig. S6g are wrong. The labels of the axes don't make sense, and the legends of the cuboid and sphere groups on the bottom left panel seems swapped. Please correct this figure.

We have improved the labels for this figure, the text in the figure caption and fixed the swapped legend in the bottom left panel in the revised version of the manuscript.

3. In Fig. S7g, can the authors provide some explanations why the decreasing trend for the asymmetric place representation disappeared for the cuboid-reared group for Day3Run1? This is an important measure used in many figures. However, this result raised the question of how reliable this measure is.

The focus/strength of this analysis is in showing positive correlation in representational warping (contributed by reduced pattern separation between the 2 track ends) vs. lack of positive

correlation in non-warping (negative or simply low non-significant absolute correlation value). We do not place emphasis on the difference between negative high and low correlations as observed in the cuboid-reared group across days. As an explanation for a non-significant correlation in cuboid-reared reared rats on Day3Run1, the ensemble population vector correlation values between even 'closer' locations on the track also became low in cuboid-reared rats (not significantly different from further away locations), likely indicative of an improved track discriminability with experience on the same linear track (Liu et al., 2021, Neuron; Cacucci et al., 2007, Journal of Neuroscience); this led to a non-significant correlation (and slope) between distance apart and the similarity of recruited ensembles. For this reason, we defined representational warping as a significant positive relationship between space apart and population vector correlations. Thus, a non-significant (as in Day3Run1) or negative relationship is not indicative of warping (significant increase in population vector similarity with distance). We added this explanation to the revised figure legend and methods as recommended by the reviewer and acknowledged that low negative correlations in cuboid-reared rats are expected in novel/not very familiar environments.

4. To compare the pattern separation of two tracks, instead of randomly shuffling cell IDs (Methods, page 54), a better way could be swapping the rate maps of the same cells across two tracks (Tirole et al., 2022).

Our goal was to compute and quantify these shared and non-shared (i.e., the ability to pattern separate) representations across tracks, and the impact of age or rearing environment on these representations. We show that both age and rearing environment impacts these population vector correlations (used to study pattern separation) and these are contributed by similar rate-representations across tracks (revised Fig. 5b, c). Further, we show that multiple individual neuronal and ensemble properties of the pre-experience sleep impacts pattern separation (revised Fig. 6f). While our results hold regardless of a shuffle, we computed a shuffle to estimate the population vector correlations for a set of random neurons to indicate where our results fall relative to it. For this purpose, we avoided the track ID shuffle for the following reasons: We compare the place map population vector correlations across tracks (Liu et al., 2021, Neuron), to study if there are shared representations of the two tracks, which we show are contributed by similar rate representations of the same neurons across the multiple tracks (Fig. 5c). In this scenario, swapping a place map of a cell is likely to have limited impact on the correlations. Therefore, we performed the cell ID shuffle to provide an estimate of a potential baseline correlation across random sets of neurons to put these population vector correlations in perspective by shuffling the similar rate representations across tracks. Tirole et al. (2022, ELife) also did not use a place map swap shuffle to quantify pattern separation (i.e., the population vector correlations across tracks) possibly due to the same reason. When studying rate replay, they used this shuffle to test if decoded replay that is 'highly' rate-specific to a particular track is observable by swapping the same cells across the two tracks before decoding and if this is coincident with trajectory replay.

We provided further explanation specifically by qualifying the usage of a shuffle only for provision of an estimate for pattern separation across a random set of neurons in the revised manuscript, in line with the recommendation of the reviewer.

5. In Fig. S3f (bottom right panels), can the authors provide some explanations why the stripes for the decoded locations were not located at the end of the track (0 or 1), which seems to be

against with their other measures of representational warping. In addition, similar stripes seem appear for the cuboid-reared group as well.

The goal of this analysis was to uncover any short-timescale (across consecutive 20 ms time bins) spatio-temporal relationships in the averaged sleep activity patterns in all frames, and if these exist, study the impact of sphere-rearing on these to test the hypothesis of ‘track-end swapping’ in sphere-reared rats. We observe that adjacent locations (manifesting as an increased probability around the diagonal of the matrix are more likely to be depicted across consecutive time bins in the sleeps of cuboid-reared rats, and there is a deficit in depicting the track ends in sphere-reared rats (the red-arrow indicates the flip from one track end to the other). We observe that the decoded population activity tapers at the absolute edge (0 or 1) like most place fields tapering at the absolute end of the track, therefore the peak of decoding is near the edge of the last bins (where the red arrow points to) instead of the absolute end (please note that we did not exclude those bins for quantification, and the results remain regardless). Regarding the underlying explanation of ‘stripes’ observed around some locations on the track in both groups: 1) This is likely due to variation in the average decoding probability of locations across the track during sleep (Farooq et al., 2019, Neuron). If certain locations are more regularly/highly depicted, they are likely to have a higher relationship with a higher number of other locations. We observe this in both cuboid and sphere-reared rats; 2) In addition, locations nearby on the track (e.g., track middle vs. track end, which are more nearby compared to the two track ends) are likely to have a higher temporal relationship with each other than locations further apart, thus they contribute to this appearance. To quantify any potential impact of this effect on our results, we conducted a control analysis, comparing the decoding specificity index of the middle (15 cm about the middle of the track) with the ends (15 cm at track end) in the cuboid- and sphere-reared rats, targeting the region referred to by the reviewer. We found that, unlike the differences in specificity of the track ends, this is not significantly different across the two groups. We share this result in Supplementary Fig. 4d as a control analysis for our results on track ends. We also provide further explanation in the revised manuscript as recommended by the reviewer.

6. Fig. 4f, “number of clusters during Day1 sleep sessions”, should it be “number of clusters during Pre-Day1Run1 sleep sessions”?

For the discrete clustering analysis, we lacked pre/post resolution on Day1, as we grouped within day sleep sessions (n=10 sleeps) and analyzed them together. We provided further details regarding this point in the revised methods section. We also added the statistical comparison of Pre-Day1Run1 sleep alone across groups to the revised manuscript (Pre-Day1Run1 sleep number of clusters, cuboid>sphere, medians: 18 vs. 13, $p<0.05$, rank-sum test).

7. Can the author provide some explanations for the difference in number of clusters shown in Fig. 6b middle panel (not significant) versus in Fig. 4f (significantly different)?

On Day1 of the experiment, the rats gained experience of linearity in two Run sessions (Day1Run1 and Day1Run2). Pre-Day1Run1 sleep and Post-Day1Run1 sleep flanked the Day1Run1, while Post-Day1Run1 sleep preceded Day1Run2. Prior experience with geometric linearity reduced the more sensitive metric, the ensemble similarity scores across sleep frame pairs and the proportion of significant frame pairs computed by using this metric in the sphere-

reared rats (revised Fig. 4b). The discrete clusters were computed based on this metric. By Day2 of the experiment, the difference across groups in the discrete number of sleep clusters was not significant anymore in the Pre-Day2Run1 sleep (revised Fig. 6b, middle panel; while overall the median number of clusters were still lower in sphere-reared rats) or Day2 sleep sessions in general, unlike in Day 1 (revised Fig. 4f). However, various properties of the frame pairs and detected clusters remained worse in the sphere-reared rats on Day2, including a higher proportion of significant frame pairs based on the ensemble similarity scores (revised Fig. 6b, left). The quality of these clusters was worse as assessed by within versus across cluster correlations of frame-pairs (revised Supplementary Fig. 6e) and the recruitment of these clusters to separately represent multiple, distinct experiences was impaired (revised Fig. 6b, right). While we did not record overnight activity from neuronal ensembles, overnight sleep following Day1Run2 likely played a role in improvement in the repertoire. While the focus of revised figures 5 and 6 was a comparison of cuboid- and sphere-reared rats on Day2 of the experiment, we explicitly studied the impact of experience across the multiple days in revised Figures 7 and 8 of our paper. We provided further explanation of this point in the revised manuscript as recommended by the reviewer.

8. Page 5, “~ 90 min” should be “~ 90 mins”.

We changed ‘min’ to ‘mins’ throughout the manuscript as recommended by the reviewer.

Reviewer #3 (Remarks to the Author):

In this manuscript, Farooq and Dragoi examined differences between rats that were reared from birth in spherical and in cuboid environments in terms of CA1 spatiotemporal representations during both wake and sleep. Similar to cuboid-reared rats, sphere-reared rats exhibited place cell activity and theta sequences during navigation, as well as place cell sequence preplay, replay and the enhancement of place cell sequence replay by experience, during sleep. However, neural codes relatively lacked complexity/specificity in sphere-reared rats, with cells showing more similar encoding of different linear tracks and of the two halves of a linear track, as well as a poorer repertoire of preconfigured neuronal patterns during sleep. These differences faded away following four days of training in the linear tracks.

This is a comprehensive study that addresses various facets of CA1 spatiotemporal encodings during wake and sleep for rats reared in spheric vs cuboid environments. I have a few comments related to the presentation of results and some potentially lacking elements of discussion.

_ In general, I found the result section not always easy to read, partly due to a lack of concision and a non-straightforward organization of analyses.

We thank the reviewer for their review and constructive feedback which improved the manuscript. In line with the reviewer’s recommendation, we have now reorganized the figures and results, attempting to make the presentation more concise as well as expanding on the less discussed points listed below by the reviewer and other reviewers.

For instance, an interesting finding is that sphere-reared rats showed symmetrical encoding of the two halves of a linear track. In my view, the presentation of the results related to this point could be more straightforward.

Several population analyses with some degree of redundancy are first presented. Not only understanding each of these analyses requires some effort, but it is also not straightforward to figure out their combined meanings. With some thinking (largely based on Supplementary Fig. 3 and Supplementary Fig. 4), one can infer that some of the cells in sphere-reared rats should have the same 2 firing fields, equidistant to track ends, for both travel directions. This is only partially mentioned in the text, in the last sentence of the 3rd paragraph of the section.

I think the section would be easier to read if the authors first showed the rate maps of a few cell/symmetric cell examples, each running direction shown separately, and the respective proportion of various cell representations (similar to Suppl Fig 4c), and then presented the population analyses which are relatively simple to interpret from a higher proportion of symmetric cells in sphere-reared than cuboid-reared rats.

We agree with the reviewer that having the single-cell/rate map analysis (previously in Supplementary Fig. 4c) in the main figures will be beneficial for the reader. Based on the observations of the reviewer, we separately analyzed each run direction and identified the proportion of rate maps characterized as each rate map type (modifying and improving the criteria for detecting each cell type based on place cells in adult cuboid-reared rats and focusing on cells near track ends). We included this analysis in the main figure (please note that the single rate map examples were separated by running direction, i.e., separate place rate maps were computed for each run direction in revised Fig. 3f, and renamed to place 'rate map' to elucidate this point). Since sphere-reared rats had a higher proportion of symmetric rate maps, we computed the proportions of single-direction rate maps exhibiting symmetry for both run directions in sphere-reared rats. This will determine if these properties were a characteristic of the single 'cell' or an instantiation at the single-cell level of ongoing ensemble network dynamics. The proportion of cells exhibiting symmetry for both run directions out of cells exhibiting symmetry for at least one run direction was low (9.2%) in sphere-reared rats (and not significantly different from cuboid-reared rats, $p > 0.05$, Z-test for two-proportions). This indicates that the representational warping might be a property of the neural network/neuronal ensemble phenomenon (as shown in the network manifold) rather than a property of individual cells within the network. We provide these explanations in the manuscript in line with the reviewer's recommendation to increase emphasis on these results.

We have reorganized the figures in line with the reviewer's recommendations to aid readability. We attempted to reorganize the figure such that the single rate map symmetry results (which had been conducted for each run direction separately) preceded the track end similarity analysis in Fig. 3a, however, 1) it was very challenging to motivate these individual rate maps' symmetry results first without presenting the population analysis on track end similarity (which is the most marked effect here, as pointed out by the reviewer and seemed to logically flow from our prior discussion on 'linearity') and warping; and 2) as we share in the previous paragraph, our results indicate that this phenomenon at the single-cell level is likely an instantiation of the ongoing network dynamics, rather than a single cell, due to the limited overlap of these rate map types across the two run directions.

Therefore, we start this figure with analysis of similarity in representation of the two track ends in sphere-reared rats (see revised Fig. 3a-b), then show that the similarity extends beyond the track ends as similarity of ensembles increases with distance of equidistant points (see revised Fig. 3d-e), and now dedicate space in the main figure to presenting and discussing an instantiation of this phenomenon at the single rate map level as well (the single rate map results which were separated by running direction, i.e., separate place rate maps were computed for each run direction shown in revised Fig. 3f, and is renamed to 'rate map' to elucidate this point) as recommended by the reviewer. We hope the current layout will aid readability of the manuscript.

As pointed out by the reviewer, in this figure and others, we provide multiple lines of evidence/analyses with occasionally a degree of overlapping emphases to strengthen our case. However, each of these analyses provide additional supporting information for the studied phenomena; for example, revised Fig. 3a, b demonstrates a higher similarity of neuronal ensemble representations of two track ends in sphere-reared rats, while revised Fig. 3c-e show that the similarity of neuronal ensembles increases with distance between symmetric location on the track in sphere-reared rats. Similarly, removing time bins from frames and studying differences in p/replay proportions across groups investigated the impact of frame starts and/or ends on occurrence of complete sequences shown in revised Fig. 3g, left, while the T-1 contingency analysis studied short-timescale spatio-temporal interactions in all the sleep frames in revised Fig. 3g, right. We have specifically focused on working towards emphasizing the distinctness of each analysis in the revised manuscript as recommended by the reviewer.

_ In several instances, the authors put forward the idea of a representational 'warping' of linear space in sphere-reared rats. While it is not an inaccurate description of the results in Fig 3c and d, it might be somewhat misleading conceptually as the reward/geometrical symmetry of the two halves of the track might likely be the critical factor of the apparent warping (representation might not be 'warped' in a linear track without rewards/endings). As is, some readers might be misled into thinking that rearing in spheric environments generally bias the network to generate warped representations of linear space.

We agree with the reviewer here that the presence of track ends (the track being finite) might contribute to the representational warping. We now further discuss this in the revised discussion section, that the warping might be contributed by the presence of track ends and a pattern separation deficit between them. We acknowledge the reviewer's point that an important future direction would be to determine if on an 'infinite' track without ends (like a running belt for instance), this phenomenon is still observed, to delineate these possibilities. To test the impact of approaching reward, we analyzed and compared the 'away' trajectories from reward, which showed similarly higher population vector correlations in the sphere-reared rats (presented in revised Supplementary Fig. 3a). These results indicate that approach to reward itself is not the sole determinant of the observed representational similarity of track ends.

Potential mechanisms for the symmetric encoding of track ends and for the larger proportion of such cells in sphere-reared rats is obscure or lacking in the discussion. Cells symmetrically encoding track ends are somewhat reminiscent of landmark/boundary/object-vector cells (Lever

et al. 2009, Deshmukh & Knierim 2013, Geiller et al. 2017, Hoydal et al. 2019), which tend to fade away with learning of environments through competitive learning (Kim et al. 2020).

One potential mechanism for representational warping which we discussed is the similarity of track ends together (lack of pattern separation between them) with a realignment of the path integration signal to each of the track ends. We agree with the reviewer, the similarity of representation of track ends (and other locations) might be due to landmark/boundary/object-vector cells, as observed in the upstream dentate gyrus in Kim et al., 2020, Nature Communications. We discussed all these papers in the revised manuscript as potential mechanisms for similar encoding of track ends and representational warping in addition to our existing references of path integration realignment as recommended by the reviewer.

REVIEWERS' COMMENTS

Reviewer #1 (Remarks to the Author):

The author responses have addressed my concerns. This is a great manuscript.

Reviewer #2 (Remarks to the Author):

The authors have addressed all the points I raised in my original review with the inclusion of extensive new analysis and discussion. I found that the manuscript has been considerably improved and fully support its publication. This will be an important contribution to the field.

Reviewer #3 (Remarks to the Author):

My comments were adequately addressed. I have only a few additional minor comments:

_ Please specify in the text or figure legend that the cell rate map examples of Fig.3f are implemented for one (or both) travel direction.

_ p11 line8: Fig3f —> Fig3g

Reviewer #3 (Remarks on code availability):

These are simple straightforward codes.

We thank the 3 reviewers for evaluating our manuscript and for their constructive, positive feedback, which has substantially improved our manuscript. Please find below our comments in response to each point the reviewers raised in italics.

REVIEWERS' COMMENTS

Reviewer #1 (Remarks to the Author):

The author responses have addressed my concerns. This is a great manuscript.

We thank the reviewer for their feedback and recognizing the impact of the manuscript.

Reviewer #2 (Remarks to the Author):

The authors have addressed all the points I raised in my original review with the inclusion of extensive new analysis and discussion. I found that the manuscript has been considerably improved and fully support its publication. This will be an important contribution to the field.

We thank the reviewer for their feedback and recognizing the impact of the manuscript.

Reviewer #3 (Remarks to the Author):

My comments were adequately addressed. I have only a few additional minor comments:

We thank the reviewer for their feedback. We have addressed their minor comments below.

_ Please specify in the text or figure legend that the cell rate map examples of Fig.3f are implemented for one (or both) travel direction.

We addressed this comment in the revised figure legend and stated that these examples were for one run direction.

_ p11 line8: Fig3f → Fig3g

We corrected this figure reference in the revised version of the manuscript.

Reviewer #3 (Remarks on code availability):

These are simple straightforward codes.

We included a link to these codes in the manuscript.